# Methionine Sulfoxide Reductases Contribute to Anaerobic Fermentative Metabolism in *Bacillus cereus*

**DOI:** 10.3390/antiox10050819

**Published:** 2021-05-20

**Authors:** Catherine Duport, Jean-Paul Madeira, Mahsa Farjad, Béatrice Alpha-Bazin, Jean Armengaud

**Affiliations:** 1Département de Biologie, Avignon Université, INRAE, UMR SQPOV, F-84914 Avignon, France; jean-paul.madeira@inrae.fr (J.-P.M.); mahsa.farjad@inrae.fr (M.F.); 2Département Médicaments et Technologies pour la Santé (DMTS), Université Paris-Saclay, CEA, INRAE, SPI, F-30200 Bagnols-sur-Cèze, France; beatrice.alpha-bazin@cea.fr (B.A.-B.); jean.armengaud@cea.fr (J.A.)

**Keywords:** methionine oxidation, methionine sulfoxide reductase, anaerobiosis, *Bacillus cereus*

## Abstract

Reversible oxidation of methionine to methionine sulfoxide (Met(O)) is a common posttranslational modification occurring on proteins in all organisms under oxic conditions. Protein-bound Met(O) is reduced by methionine sulfoxide reductases, which thus play a significant antioxidant role. The facultative anaerobe *Bacillus cereus* produces two methionine sulfoxide reductases: MsrA and MsrAB. MsrAB has been shown to play a crucial physiological role under oxic conditions, but little is known about the role of MsrA. Here, we examined the antioxidant role of both MsrAB and MrsA under fermentative anoxic conditions, which are generally reported to elicit little endogenous oxidant stress. We created single- and double-mutant Δ*msr* strains. Compared to the wild-type and Δ*msrAB* mutant, single- (Δ*msrA*) and double- (Δ*msrA*Δ*msrAB*) mutants accumulated higher levels of Met(O) proteins, and their cellular and extracellular Met(O) proteomes were altered. The growth capacity and motility of mutant strains was limited, and their energy metabolism was altered. MsrA therefore appears to play a major physiological role compared to MsrAB, placing methionine sulfoxides at the center of the *B. cereus* antioxidant system under anoxic fermentative conditions.

## 1. Introduction

*Bacillus cereus* is a notorious food-borne gram-positive pathogen that can adapt to various oxygenation conditions encountered in the environment, in foods and in the human intestine [1]. Under anaerobiosis and in the absence of a final electron acceptor such as nitrate, *B. cereus* sustains its growth by producing ATP through mixed acid fermentation. This process generates lactate as the main metabolite, along with smaller amounts of acetate, formate, succinate, and ethanol [2,3,4]. Traditionally, fermentative conditions are not considered to produce reactive oxygen species (ROS), due to the decreased oxidative phosphorylation taking place. However, experimental evidence suggests that an antioxidant response nevertheless occurs in fermenting *B. cereus* cells [5,6]. This response may be due to *B. cereus* undergoing transient secondary oxidative stress upon exposure to anaerobic fermentative conditions, through nitric oxide (NO) production [7], as a result of the action of nitric oxide synthase on arginine [8,9,10].

The sulfur-containing methionine residues in proteins are especially sensitive to ROS-mediated oxidation [11]. Oxidation of methionine results in two diastereomic forms of methionine sulfoxide (Met(O)): methionine-S-sulfoxide (Met-S-O) and methionine-R-sulfoxide (Met-R-O). The reaction is reversible thanks to the action of methionine sulfoxide reductase (Msr) [12]. Four types of Msr have been identified that reduce Met(O) residues to their functional form [13]. The main types are the thiol-oxidoreductases MsrA and MsrB, which react specifically with diastereomers displaying the S- and R-configurations at the sulfur atom, respectively [14,15]. The third and fourth types of Msr belong to the molybdenum-containing enzyme families. The third type of Msr, periplasmic methionine sulfoxide reductase (MsrP), is present in most gram-negative bacteria and is a member of the sulfite oxidase family. MsrP is not a stereospecific protein-MetO reductase [16]. The fourth type of Msr belongs to the dimethyl sulfoxide (DMSO) reductase family. One example is the periplasmic *Rhodobacter sphaeroides* DorA DMSO reductase, which reduces DMSO and both free and protein-bound Met-S-O [13]. The primary role of the first three types of Msr is to regulate the Met(O) level in proteins; they reduce Met(O) residues more efficiently in unfolded proteins than in folded proteins [17]. The role of DorA-like enzymes in protecting proteins against oxidation remains to be validated in vivo [16]. Cyclic Msr-dependent methionine oxidation/reduction was proposed to be an important antioxidant defense mechanism in both bacteria and eukaryotic cells, playing a scavenging role for cellular ROS under normal and stressed conditions [11,17,18,19,20]. Msr can also regulate protein function by modulating specific Met(O) residues involved in protein activation or inactivation [21]. Consequently, these enzymes are implicated in a variety of biological processes, including redox signaling, cellular metabolism, and virulence. Although Msr and other antioxidants have been identified in both facultative and strict anaerobes [22,23], their role in cellular processes under anoxic conditions remains largely uncharacterized. The few studies available report results for yeast, where MsrA and MsrB were reported not to contribute to the ROS-regulated lifespan under strict anoxic conditions [24], and for the strict anaerobic bacterium *Clostridium oremlandii*, where high activity levels of the selenoprotein MsrA were detected [25]. However, no physiological role has yet been proposed for these enzymes in anoxic conditions.

Whole-genome sequence analysis of *B. cereus* ATCC 14579 revealed two genes encoding Msr. The first of these genes, *msrAB*, encodes a bifunctional cytoplasmic MsrAB enzyme, and is probably the result of gene fusion [26]. Our previous results indicated that MsrAB plays a key role in redox homeostasis in *B. cereus* under aerobic respiratory conditions, and regulates exotoxin secretion [27]. The role of MsrA has not been investigated to date. For this study, we generated single and double mutants of *msrA* and *msrAB* to investigate the antioxidant role of Msr proteins, under anaerobic fermentative growth conditions, using a proteomics approach. The results suggest that the repair of Met(O)-containing proteins contributes significantly to anaerobic fermentative growth of *B. cereus*, and demonstrate clearly distinct in vivo contributions from MsrA and MsrAB to the reduction of cellular protein-bound Met(O).

## 2. Materials and Methods

### 2.1. Strains, Media, and Growth Conditions

Wild-type *B. cereus* ATCC 14579 without its pBClin15 plasmid [28], the single mutants Δ*msrA*, Δ*msrAB* [27], and the Δ*msrA*Δ*msrAB* double mutant strains were grown either in Luria broth (LB) or MOD medium supplemented with 30 mM glucose (MODG) as the carbon source, as previously described [3]. Controlled anaerobic batch cultures were performed at 37 °C in 2 L bioreactors (Discovery 100, Iceltech, France) containing 1.5 L MODG medium. pH was maintained at 7.2 by automatic addition of 5 M KOH, and constant shaking at 300 rpm. Anaerobiosis (pO_2_ = 0%) was obtained by continuously flushing the medium with pure nitrogen gas (20 mL/h) that had been passed through a Hungate column. The inoculum was a 100-mL sample of an exponential anaerobic culture (250 mL Erlenmeyer flasks fitted with open-top caps and rubber septa) harvested by centrifugation, washed twice and diluted in fresh MODG medium to obtain an initial optical density at 600 nm of 0.02. Three biological replicates were analyzed for each strain.

### 2.2. Analytical Procedures

The growth of *B. cereus* WT and mutant strains was monitored spectrophotometrically at 600 nm. The maximal specific growth rate (µ_max_) was determined by applying the modified Gompertz equation [29]. Culture samples (250 mL) were harvested anaerobically during the early exponential (EE, µ= µ_max_), late exponential (LE), and stationary (S) growth phases and treated in an anaerobic chamber. Cells and culture supernatants were separated by centrifugation (10,000× *g*, 10 min, 4 °C). Cells were suspended in PBS buffer, and stored at −20 °C. Supernatants were filtered as previously described [6], aliquoted in small volumes, and stored frozen at −20 °C until analysis.

Glucose, lactate, ethanol, formate, acetate, and succinate concentrations were determined in filtered supernatants using Enzytec Fluid kits (R-Biofarm, Saint-Didier-au-Mont-d’Or, France) according to the manufacturer’s protocol.

Cellular and extracellular proteins were extracted from cell pellets and filtered supernatant samples as previously described [27,30]. All manipulations were performed in an anaerobic chamber to avoid artefactual oxidation.

### 2.3. Relative Quantification of msrA and msrAB Gene Expression

Real time quantitative RT-PCR experiments were performed as described previously [27]. Briefly, real-time RT-PCR was performed using the iScript™ One-Step RT-PCR kit with SYBR^®^ Green following the manufacturer’s protocol (Biorad, Marne-La-Coquette, France), with 10 ng of total RNA as a template. Total RNA was extracted from *B. cereus* cells at EE, LE and S growth phases by using TRI Reagent RNA extraction solution as recommended by the manufacturer (Ambion, Thermo Fischer Scientific, Illkirch, France). The Agilent 2100 Bioanalyzer system was used to characterize and quantify total RNA. The mRNA level changes for each gene were normalized to the RNA level for the *ssu* gene, encoding 16S RNA, and quantified by the 2−ΔΔCT method. The specific primer pairs used in these experiments were: 5′-TTCTGGTACACAGGTGGTC-3′ and 5′-AAAGCGTCCACTCTGCTCAA-3′ for *msrAB* (BC_5436, NC_004722.1); 5′-TCCAACTGATGATGGCGGAC and 5′-TCACGCCCAGATTCTTTTTGC-3′ for *msrA* (BC_1774, NC_004722.1). Data from two technical replicates, for each of the three biological replicates, were used to conduct statistical analysis.

### 2.4. Construction of msrA Mutant Strains

The *msrA* gene (BC 1774) was disrupted in *B. cereus* ATCC 14579 by allelic exchange with a spectinomycin resistance cassette (Sp^R^, [31]), and in the *B. cereus* Δ*msrAB* strain by exchange with a kanamycin resistance cassette (Km^R^), as described by Arnaud et al. ([32]). Briefly, a 949-bp DNA fragment encompassing the *msrA* gene was obtained by PCR using the following primers: 5′GAATTCGCTTAGGTGAAGTAGAAGACATTG-3′ (with an EcoRI site at the 5′end) and 5′AGATCTGTATATAAGATGGACAATTAAACAAAG-3′ (with a BglII site at the 5′ end). The PCR product was cloned into pCRXL-TOPO (Invitrogen, Thermo Fischer Scientific, Illkirch, France), and the recombinant plasmid (pCRXL-TOPO:msrA) was digested with XmnI. The 1.5-kb Sp^R^ cassette was isolated from pDIA [2] by digestion with SmaI, and the 1.5-kb Km^R^ cassette was isolated from pDG789 [33] by digestion with SmaI and StuI. The resistance cassettes were ligated into the XmnI-digested pCRXL-TOPO:msrA vector. The resulting plasmids were subsequently digested with EcoRI and BglII to extract the msrA::Spr^R^ and msrA::Km^R^ fragments. These fragments were cloned into pMAD precut with the same restriction enzymes. The pMAD-*msrA*::Spr^R^ plasmid was introduced into *B. cereus* WT and the pMAD-*msrA*::Km^R^ was introduced into the Δ*msrAB* mutant by electroporation. The *msrA* gene was deleted in both strains by a double crossover event. Chromosomal allele exchanges were confirmed by PCR with oligonucleotide primers located upstream and downstream of the DNA regions used for allelic exchange. For some experiments, the *msrA* gene was complemented in trans by cutting the pCRXL-TOPO:*msrA* plasmid with EcoRI and *Bgl*II and ligating the product to similarly digested pHT304 [34]. The integrity of the insert in the recombinant vector was verified by sequencing, and the vector was then used to transform the *B. cereus* mutant strains.

### 2.5. Proteomics Analysis

Cellular and extracellular proteins from the 36 samples (3 biological replicates × 3 time-points × 4 strains) of WT, Δ*msrA*, Δ*msrAB* and Δ*msrA*Δ*msrAB* strains were subjected to a short electrophoretic migration (about 3 min) on NuPAGE 4–12% Bis-Tris gels (Invitrogen, Thermo Fischer Scientific, Illkirch, France) using NuPAGE MES supplemented with NuPAGE antioxidant as running buffer [35]. Proteins were then submitted to in gel proteolysis with sequencing-grade trypsin (Roche, Sigma Aldrich, Saint Quentin Fallavier, France) according to the ProteaseMAX protocol (Promega, Madison, WI, USA) [36,37]. Samples were analyzed by Liquid Chromatography with Tandem Mass spectrometry (nanoLC-MS/MS) using an LTQ-Orbitrap XL hybrid mass spectrometer (Thermo Fisher Scientific, Illkirch, France) coupled to an Ultimate 3000 nRSLC system (Dionex, Thermo Fisher Scientific, Illkirch, France). NanoLC-MS/MS analysis was performed as previously described [27,38]. Briefly, peptides were resolved on a Dionex nanoscale Acclaim Pepmap100 C18 capillary column (3 μm bead size, 100 Å pore size, 75 μm i.d. × 15 cm) at a flow rate of 0.3 μL/min over 90 min, applying a gradient from 5 to 60% solvent B (0.01% HCOOH/100% CH_3_CN) and, over 180 min, applying a gradient from 5 to 50% solvent B. Solvent A was 0.01% HCOOH/100% H_2_O. Tryptic MS/MS spectra were searched against an in-house *B. cereus* ATCC 14579 database using the MASCOT Daemon search engine with the following parameters: 5 ppm peptide tolerance, 0.5 Da fragment ion tolerance, 2^+^ or 3^+^ peptide charge, a maximum of two missed cleavages, carbamidomethylation of cysteine (+57.0215) as fixed modification, and oxidation of Met (+15.5949) as a variable modification. Only peptides identified with a *p*-value < 0.05 in homology threshold mode, and proteins identified by at least two distinct peptides, were retained upon parsing with IRMa software v1.3.1 [39]. The false positive rate determined from the corresponding decoy database was estimated to be less than 1%.

Changes in protein abundance and Met(O)peptide levels between WT and mutant strains at the different time-points were analyzed using the Bioconductor DEP package (version 1.12.0), and R version 4.0.2 [40]. Met(O)peptides were first filtered to ensure two valid identifications in at least two biological replicates. Relative Met(O)peptide abundance levels were calculated based on the Met(O) spectral count normalized with respect to the total number of spectra. Missing values were imputed using random draws from a Gaussian distribution centered around a minimal value. Relative protein abundances were calculated based on spectral counts after correction and normalization by variance stabilizing (vsn), using the Limma package. Significant changes were selected where the adjusted *p*-value was less than 0.05 and the |fold-change| ≥1.5.

Mass spectrometry associated proteomics data have been deposited to the ProteomeXchange Consortium via the PRIDE partner repository under dataset identifiers PXD024888 for the cellular proteome of *B. cereus* ATCC 14579, PXD024927 for the cellular proteome of the Δ*msrAB* mutant, PXD024850 for the cellular proteome of Δ*msrA* mutant, PXD024849 for the cellular proteome of Δ*msrAB*Δ*msrA* mutant, PXD024702 for the exoproteome of *B. cereus* ATCC 14579, PXD024714 for the exoproteome of Δ*msrAB* mutant, PXD024847 for the exoproteome of Δ*msrA* mutant, and PXD024848 for the exoproteome of the Δ*msrAB*Δ*msrA* mutant.

### 2.6. Motility Assays

The *B. cereus* WT and mutant strains were grown overnight and surface-spotted (5 µL) in the center of Tryptone-NaCl plates (1% tryptone, 0.5% NaCl) semi-solidified with 0.25% and 0.7% agar to assess swimming and swarming motility, respectively [41]. Motility diameters (indicative of migration of the bacteria from the center to the periphery of the plate) were measured after 72 h incubation at 37 °C in AnaeroPack™ Jars (Thermo Fisher, Scientific, Illkirch, France). Biological triplicates were performed for all experiments.

### 2.7. Statistical Analyses

Data from three biological replicates were pooled for statistical analyses. Comparisons among multiple groups were analyzed by analysis of variance (ANOVA) followed by Tukey’s post hoc analysis (qRT-PCR and Met(O) quantification experiments). Changes in motility and metabolite production were evaluated using the Student’s t-test. Statistical analyses were performed using XLSTAT software (version 2021.1, Addinsoft, Paris, France). *p*-values ≤ 0.05 were considered significant.

## 3. Results

### 3.1. Expression Patterns for B. cereus msrA and msrAB RNA and Proteins under Fermentative Anaerobic Conditions

In silico analysis of the *B. cereus* ATCC 14579 genome identified two genes encoding methionine sulfoxide reductase. The first gene, *msrAB* (BC_5436), has been described elsewhere and encodes a cytoplasmic MsrAB protein [27]. The second gene (BC_1774) was annotated as *msrA*. It encodes a predicted cytoplasmic MsrA protein consisting of 178 amino acids, with a theoretical molecular mass of 20475 Da. This protein contains an MsrA domain that shares 64% sequence identity to the MsrA domain of the bifunctional MsrAB protein.

We compared the mRNA levels of *msrA* and *msrAB* during the EE, LE, and S growth phases under anoxic conditions. Figure 1 shows that maximum mRNA expression levels for *msrA* and *msrAB* were reached during the LE growth phase and, in contrast to *msrAB*, the mRNA level for *msrA* was higher during the EE growth phase than during the S growth phase (*p* < 0.05). The kinetics of *msrAB* expression under anaerobiosis was distinct from that described previously under aerobiosis [27]. Both *msrA* and *msrAB* mRNA levels increased under aerobiosis during active growth, and remained high during the stationary growth phase (Appendix A). This expression profile suggests that the growth-phase-dependent expression of *msr* genes is influenced by oxygenation conditions.

We next performed high-resolution tandem mass spectrometry analyses to detect MsrA and MsrAB in the *B. cereus* anoxic cellular proteome. Only one peptide was assigned to each of the two Msr in this analysis (data not shown). The same analysis of an oxic cellular proteome detected two and five peptides for MsrA and MsrAB, respectively [27]. No Msr-peptide was detected in *B. cereus* exoproteome whatever the oxygenation conditions, indicating that both MsrA and MsrAB are cytoplasmic proteins. These results suggest that both Msr proteins are expressed at low abundance, and that their levels are decreased when cells are grown in anaerobic conditions compared to aerobic growth conditions.

### 3.2. Effect of msrA and msrAB Mutations on Anaerobic Fermentative Metabolism

We previously reported the construction of a Δ*msrAB* mutant strain [27]. For the present study, we disrupted the *msrA* gene in WT and *msrAB* mutant strains to create a single Δ*msrA* mutant and a double Δ*msrA*Δ*msrAB* mutant, respectively. Quantitative RT-PCR analyses confirmed that both mutant strains lack *msrA* expression. Expression was restored by complementation of these strains with a plasmid carrying the *msrA* gene (pHT304-*msrA*) (data not shown).

We first compared the growth features of the Δ*msrA*, Δ*msrAB* and double Δ*msrA*Δ*msrAB* mutants to those of the parental ATCC 14579 wild-type strain (Figure 2). A similar maximal specific growth rate (µ_max_) was measured for all four strains (Table 1).

However, Δ*msrA*Δ*msrAB* and Δ*msrA* mutants reached the stationary growth phase at a lower final biomass than the Δ*msrAB* mutant and WT strains (1.3 and 2-fold decrease, respectively, Figure 2, Table 1). Interestingly, the final biomass recorded for the Δ*msrA* mutant was significantly higher than that for the Δ*msrA*Δ*msrAB* double mutant. The highest specific glucose uptake rate (q_glucose_) was measured for the Δ*msrA*Δ*msrAB* double mutant, indicating that glucose supports higher glycolytic fluxes in the absence of Msr. As with the WT strain, glucose was mainly metabolized into lactate by mutant strains, and no difference in terms of lactate secretion yield was observed (Table 1). However, the Δ*msrA* and Δ*msrA*Δ*msrAB* mutants produced more acetate and smaller amounts of succinate than the Δ*msrAB* mutant and WT strains. The Δ*msrA*Δ*msrAB* double mutant accumulated higher levels of acetate in the growth medium than the Δ*msrA* single mutant. Taken together, these data suggest that MsrA mediates a much more significant regulatory effect on fermentative metabolism than MsrAB, and that, together, the two proteins may contribute to the regulation of carbon flow at the pyruvate node.

### 3.3. Met(O) Accumulation in msr Mutant Proteomes

Since the lack of Msr activity makes cellular proteome and exoproteome more susceptible to oxidation [27], we compared the total Met(O) peptide content of cellular (Appendix A) and extracellular (Appendix A) protein samples harvested at EE, LE and S growth phase from cultures of *msr* mutants and WT strains (Figure 2). The Met(O) peptide content was estimated as a percentage of the total number of peptides identified in each of the three biological replicates obtained from each growth phase. The data showed no significant changes in intracellular Met(O) peptide content as growth progressed for any strain (Figure 3a). However, higher Met(O) levels were detected in Δ*msrA* and Δ*msrA*Δ*msrAB* mutant strains compared to Δ*msrAB* mutant and WT strains regardless of the growth phase, and in the Δ*msr*Δ*AmsrAB* double mutant compared to the *msrA* single mutant at the EE and LE growth phases. Met(O) peptide content accounted for 8% of all peptides expressed by the Δ*msrA*Δ*msrAB* mutant at the beginning of the exponential growth phase (EE). No significant differences in extracellular Met(O) were found between the mutant and the parental strains (Figure 3b). Met(O) accounted for up to 13% of the total extracellular peptide count. Taken together, these data show that MsrA is a major contributor to the regulation of cellular proteome-wide methionine oxidation under anaerobiosis, whereas MsrAB appears to play a minor role in this process.

### 3.4. Identification of Putative Msr Substrates

Proteins carrying MetO residues that are differentially oxidized in *msr* mutants are likely to be substrates for Msr. To identify Msr substrates, we considered that a peptide contained Met(O) residues (Met(O) peptide) when the oxidized form was detected in a least two biological replicates at different time-points. On this basis, we identified 476 Met(O) peptides in cellular extracts (Appendix A) and 370 Met(O) peptides in extracellular extracts (Appendix A). We then examined changes in Met(O) levels in these peptides. Only peptides for which the Met(O) content changed more than 1.5-fold in at least one mutant strain compared to WT (adjusted *p*-value < 0.05) were selected. Based on this criterion, 43 Met(O)-peptides from 15 proteins were confidently identified from cellular extracts (Table 2), and 35 Met(O)-peptides from 21 proteins were identified from extracellular extracts (Table 3). To exclude any influence of bias in protein abundance on the observed differences in levels of Met(O), we performed, in parallel, a differential proteomics analysis on the 998 cellular proteins and 433 exoproteins validated by at least two peptides (Appendix A). The statistical criteria applied were the same as those used to determine the Met(O) level. This analysis revealed 43 cellular proteins (Appendix A) and 41 extracellular proteins (Appendix A) for which significant abundance changes were detected. Two cellular proteins, the glycolytic enzyme glucose 6-phosphate isomerase and the protein Gls24, were found to be increased both in terms of their Met(O) content (Table 2) and their abundance in Δ*msrA* and Δ*msrA*Δ*msrAB* mutants compared to WT (Appendix A). Both the Met(O) level and overall abundance of four extracellular proteins were decreased in mutant strains compared to WT (Table 3 and Appendix A). Among these proteins, three (flagellins FlaA, FlaB and FlaC) are components of the flagellar apparatus. Finally, out of the 15 proteins listed in Table 2, thirteen are potential substrates of Msr, whereas, out of the 21 extracellular proteins listed in Table 3, seventeen may be potential substrates of Msr.

#### 3.4.1. Putative Msr Substrates Identified in *B. cereus* Cellular Proteome

According to our criteria, no significant differences in abundance for individual protein-bound Met(O) residues were found between the Δ*msrAB* mutant and WT strains, regardless of the growth phase. In contrast, in Δ*msrA* and/or Δ*msrA*Δ*msrAB* mutants, a higher Met(O) content was detected for 32 Met residues compared to the levels present in WT (Table 2). The abundance levels for the corresponding 13 proteins were unchanged (Appendix A). Interestingly, we noted that the highest number of Met(O) level changes in Δ*msrA* and/or Δ*msrA*Δ*msrAB* mutants was detected in samples harvested during the LE growth phase. These results indicate that MsrA has a broader substrate range than MsrAB, and that the impact of MsrAB deficiency was minor in the absence of MsrA.

The translation elongation factor Ef-Tu (Tuf)—one of the most abundant proteins in cells—contains the largest number of MsrA-Met(O) targets, representing 83% of its total Met content. The chaperonin GroEL was also found to contain a high number of Met(O) susceptible to MsrA reduction (five residues), representing 25% of its Met content. The Met content of glycolytic enzymes enolase (Eno) and glyceraldehyde-3-phosphate dehydrogenase (GapA2) was 45% more oxidized in the absence of MsrA. Several other enzymes classed as involved in energy metabolism were found to be more oxidized in Δ*msrA* mutant strains. These enzymes included the fermentative enzyme alcohol dehydrogenase (AdhE), the glycolytic enzyme pyruvate kinase (Pyk), the fermentative enzyme lactate dehydrogenase (Ldh2), and the alpha subunit of ATP synthase/hydrolase (AtpA). However, in contrast to Eno and GapA2, each of these enzymes contained only one MsrA-Met(O) target, and their Met(O) residues represented no more than 14% of their total Met content. Regardless of the number of Met(O) targets, many of these proteins corresponded to key enzymes in the energy metabolism.

#### 3.4.2. Putative Msr Substrates Identified in *B. cereus* Exoproteome

Many of the proteins identified as putative Msr substrates in *B. cereus* exoproteome were typical exoproteins, including exotoxins and degradative enzymes (Table 3). Changes in their Met(O) content were observed mainly during the LE growth phase in all three mutant strains—Δ*msrA*, Δ*msrAB* and Δ*msrA*Δ*msrAB*—indicating that some of their Met(O) residues can be substrates of both MsrA and MsrAB. Degradative enzymes contained no more than two Met residues for which increasing Met(O) levels were detected. Overall, exotoxins contained more Met(O) residues than other proteins. Specifically, Hbl components contained the highest number of Met(O) residues, with 37.5 and 50% of total Met residues in the lytic component HblL1 and the binding component HblB, respectively. In conclusion, our results indicate that several virulence factors secreted by *B. cereus*, including toxins and degradative enzymes, are putative Msrsubstrates under anaerobiosis, particularly during the LE growth phase.

### 3.5. Swimming Motility of B. cereus Depends on MsrA Activity

Our proteomics results indicated that MsrA activity was essential to maintain the abundance level of flagellins (which drive *B. cereus* motility [42]). We therefore investigated swimming and swarming motility for the WT strain and compared it to that of the Δ*msrA,* Δ*msrAB* and Δ*msrA*Δ*msrAB* mutants under anaerobiosis. Data were evaluated by a Student t-test. No significant difference in cell migration capacity was observed between WT and mutant strains on media supporting swarming. In contrast, as shown in Figure 4, the Δ*msrA* and the Δ*msrA*Δ*msrAB* mutants gave rise to significantly smaller colonies (diameter 5.0 ± 0.0 mm) than those produced by the WT and Δ*msrAB* mutant strains (diameters 10.7 ± 0.6 and 10.3 ± 0.6 mm, respectively; *p*-value < 0.01). This reduced diameter suggests that *B. cereus* swimming motility was significantly reduced in the absence of MsrA.

## 4. Discussion

The aim of this study was to investigate the antioxidant role of Msr proteins under anaerobic fermentative growth conditions by a proteomics approach. MsrA and MsrAB are two methionine sulfoxide reductases predicted to reverse oxidative damage to methionine in *B. cereus*. The genes encoding these two Msr are transcribed independently, and our results show that they share similar expression patterns over the course of fermentative anaerobic growth. Thus, expression of *msr* genes is growth-phase dependent and reached its maximum during the LE growth phase under anaerobiosis; under aerobiosis, this maximum was reached during the stationary phase [27]. This expression profile suggests that Msr are required earlier when cells are grown under anaerobiosis compared to under oxic conditions, perhaps as a means to repair accumulated oxidative damage [43]. The contributions of MsrA and MsrAB to the reduction of cellular Met(O)-bound proteins were clearly distinct under fermentative anaerobic conditions. The extended substrate range of MsrA compared to MsrAB could explain the major role played by MsrA compared to MsrAB in *B. cereus* fermentative growth.

In contrast to MsrAB, a lack of MsrA significantly changed the oxidation status of Met residues in several cellular proteins including the GroEL chaperone, the elongation factor Ef-Tu, the alpha component of the F1F0-ATPase complex and key enzymes involved in central metabolic processes, which led to concomitant changes in the distribution of metabolic fluxes (Figure 5).

GroEL is a Met-rich chaperone that functions in vivo to fold newly synthetized peptides. The chaperonin activity of GroEL needs ATP [44] and, in *Escherichia coli*, depends on a Msr-mediated repair system under oxic growth conditions [45]. In *Helicobacter pylori*, the Msr-mediated repair of GroEL serves to protect catalase against oxidative damage [46]. The elongation factor Ef-Tu is a translational GTPase that plays a central role during the elongation phase of protein synthesis in bacteria and eukaryotes. It also displays diverse moonlighting functions related to bacterial pathogenesis [47]. In eukaryotes, Msr-based repair systems preserve the activity of Ef-Tu and other complexes to maintain protein biogenesis [16,17]. Under fermentative conditions and in the absence of oxidative phosphorylation, the F0F1 ATPase complex hydrolyzes ATP and plays a key role in the H^+^ transport accompanying certain secondary transporters and/or enzymes involved in anaerobic oxidation–reduction [48]. Impairment of F0F1 as a result of enhanced Met oxidation of one of its components could increase its turnover and ATP requirements, leading to perturbed fermentative metabolism (Table 1). Changes to the oxidation status of glycolytic enzymes and fermentative enzymes, and thus in their activities [49], could also contribute to perturbing fermentative metabolism. The lactate pathway is the main fermentative pathway regenerating NAD^+^ from NADH in *B. cereus* [2]. Indeed, during pyruvate oxidation, *B. cereus* produces significantly more lactate than succinate and ethanol (Table 1). Lactate and ethanol production were unchanged in the absence of MsrA, whereas succinate formation was decreased. However, due to the minor impact of the succinate pathway on NAD^+^ generation, this decrease probably has a neutral effect on the cellular redox balance. The acetate pathway does not involve redox reactions and generates additional ATP outside of glycolysis through the conversion of acetyl-CoA to acetate (Figure 5). In the absence of MsrA, acetate secretion is increased, indicating a higher carbon flow through acetate pathway, and a concomitantly higher ATP production. To sustain high carbon flow through the acetate pathway, Δ*msrA* and Δ*msrA*Δ*msrAB* mutants increase carbon flow through glycolysis, as revealed by increased glucose consumption (Table 1). Finally, when MsrA is lacking, *B. cereus* adjusts its fermentative metabolism to maintain redox balance and promote ATP synthesis, probably to sustain—for as long as possible—repairing processes that have become less efficient due to the absence of Msr, and F0F1 ATPase complex deficiency.

In terms of capacity to colonize an environment, in the absence of MsrA, *B. cereus* cells showed reduced motility. This result appears logical as the bacterial flagellum is responsible for motility [50], and synthesis of some flagellar components was observed to be reduced in the absence of MsrA, possibly as a result of an overoxidation of their Met residues [51]. Down-regulation of flagellar components in a context where ATP demand is increased may help cells to maintain growth. The flagellum is a potential virulence factor along the same lines as degradative enzymes and exotoxins. All these virulence factors contain several Met(O), are highly produced at the end of growth [52], and are exported in an unfolded form [53]. Msr are known to preferentially reduce unfolded oxidized proteins [17], suggesting that these virulence factors, especially enterotoxin Hbl, could be repaired by cytoplasmic Msr before their secretion. Whether Msr-dependent Met oxidation regulates exotoxin secretion, structure and activity remains an open question. However, as reported for other pathogens [54], it is probable that Msr contributes to *B. cereus* virulence under both anaerobiosis and aerobiosis [27].

In contrast to MsrA, a lack of MsrAB had little effect on *B. cereus* fermentative metabolism. Although it has a minor impact compared to MsrA, the role of MsrAB is, nevertheless, important as it contributes to the intracellular accumulation of Met(O) (Figure 3), and to fermentative metabolism in the absence of MsrA. This observation suggests that MsrA could have a higher affinity than MsrAB for surface-accessible Met(O) within cellular proteins, and that MsrA and MsrAB could cooperate to reduce Met(O) in some proteins. Differences in Met(O) reduction activity between MsrAB and MsrA were less marked when examining extracellular proteins, suggesting that the affinity differences between the two enzymes could be less significant when dealing with unfolded proteins.

## 5. Conclusions

Even though MsrA and MsrAB seem to be present at very low levels in anaerobic fermentative *B. cereus* cells, they are part of a highly regulated machinery controlling energy metabolism and, possibly, virulence (Figure 5). In addition, our study indicates that the oxidation of methionine residues in proteins may be an inevitable side effect of life, whether under anaerobic or aerobic conditions.

## Figures and Tables

**Figure 1 antioxidants-10-00819-f001:**
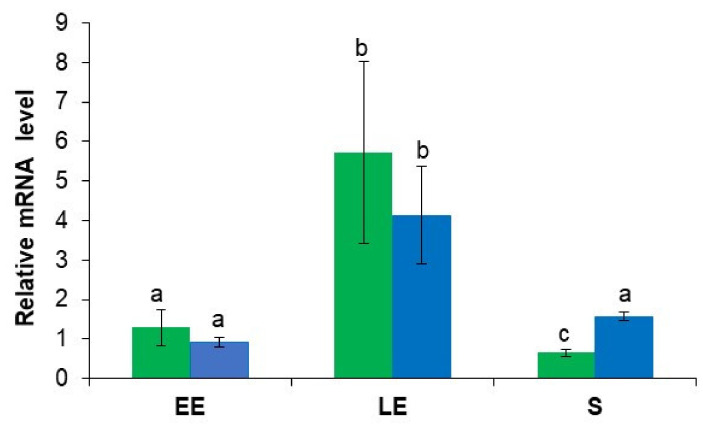
Analysis of *msrA* and *msrAB* gene expression in *B. cereus* ATCC 14579 cells grown under anaerobiosis. mRNA levels of *msrA* (green) and *msrAB* (blue) were determined at the early exponential (EE), late exponential (LE) and stationary (S) growth phases by quantitative real-time RT-PCR and normalized to *ssu* mRNA levels. Samples were harvested as indicated in Figure 2. Data correspond to the mean ± SD of six samples (two technical replicates x three biological replicates). Data denoted by a common letter are not significantly different. Data denoted by different letters indicated a significant difference (two-way ANOVA followed by Tukey’s multiple comparison post hoc analysis, *p* ≤ 0.05).

**Figure 2 antioxidants-10-00819-f002:**
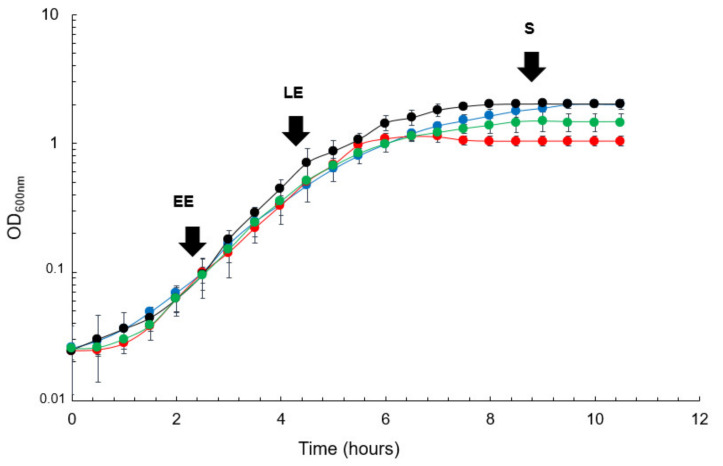
Growth curves for *B. cereus* wild-type and *msr* mutant strains under anaerobic fermentative conditions. Wild-type strain ATCC 14579 (black), Δ*msrA* (green), Δ*msrAB* (blue), and Δ*msrA*Δ*msrAB* mutants (red) were grown in MOD medium supplemented with 30 mM glucose. Data correspond to the mean ± SD of three biological replicates.

**Figure 3 antioxidants-10-00819-f003:**
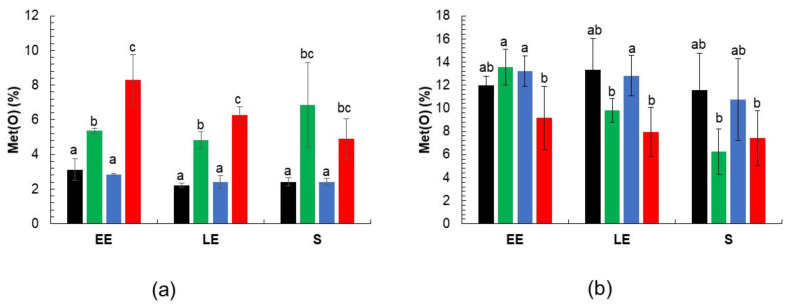
Met(O) content of *B. cereus* wild-type and *msr* mutant proteomes under anaerobic fermentative conditions. Wild-type strain ATCC 14579 (black), Δ*msrA* (green), Δ*msrAB* (blue), and Δ*msrA*Δ*msrAB* mutants (red), were grown in MOD medium supplemented with 30 mM glucose. Samples were collected at early exponential (EE), late exponential (LE), and stationary (S) growth phases, as indicated in Figure 2. The relative Met(O) content in the cellular proteome (**a**) and exoproteome (**b**) was calculated as a percentage of oxidized methionine-containing peptides with respect to the total number of methionine-containing peptides identified in samples for each growth phase. Data correspond to the mean ± SD of three biological replicates. Within each panel, data denoted by a common letter are not significantly different. Data designed by different letters indicated a significant difference (two-way ANOVA followed by Tukey’s multiple comparison post hoc analysis, *p* ≤ 0.05).

**Figure 4 antioxidants-10-00819-f004:**
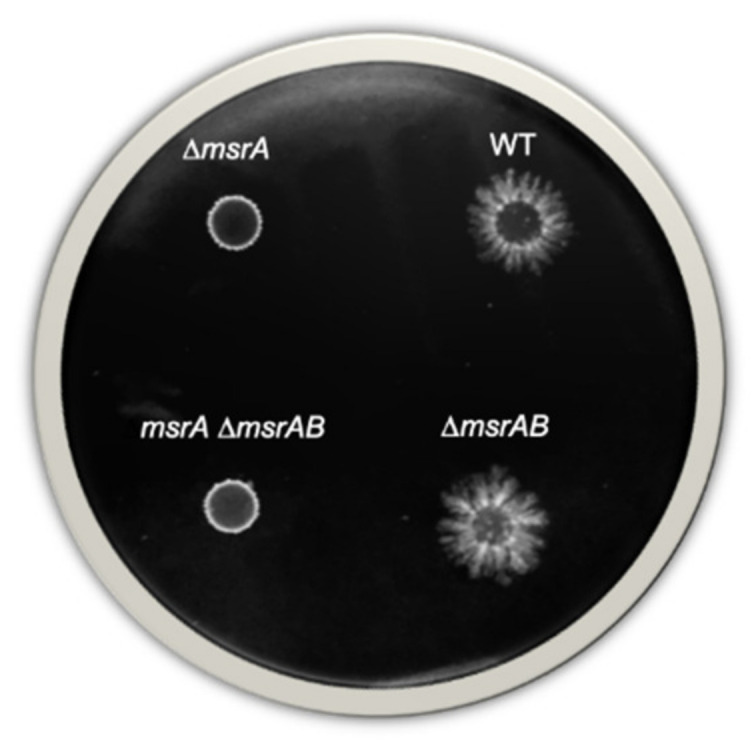
Swimming motility. Representative colonies of wild-type (WT) and Δ*msrA*, Δ*msrAB* and Δ*msrA*Δ*msrAB* mutants following growth on swimming TrA plates for 72 h under anaerobiosis.

**Figure 5 antioxidants-10-00819-f005:**
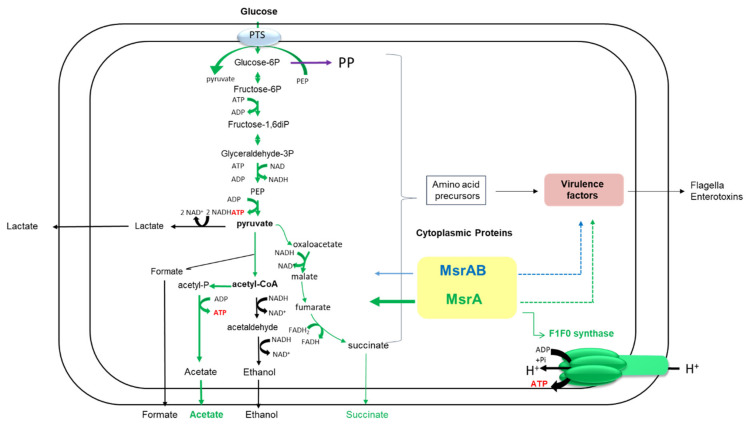
Schematic view of the roles of MsrA and MsrAB on *B. cereus* physiology under anaerobic fermentative conditions. MsrA has a more important role in fermentative metabolism, compared to MsrAB (thick green vs. thin blue solid arrows). The role of MsrA could be as important as the role of MsrAB in maintaining Met(O) levels of virulence factors, such as flagella and enterotoxins (green and blue dotted arrows). PTS, phosphotransferase system; PP, pentose phosphate pathway. Succinate is formed via the reductive TCA cycle.

**Table 1 antioxidants-10-00819-t001:** Growth parameters and end product yields obtained for anaerobic batch cultures of Δ*msrA*, Δ*msrAB* and Δ*msrA*Δ*msrAB* mutants, and their parental strain, *B. cereus* ATCC 14579 (WT).

	*B. cereus* Strains			
	WT	Δ*msrA*	Δ*msrAB*	Δ*msrA*Δ*msrAB*
µ_max_ (h^−1^)	1.0 ± 0.1 ^a^	1.0 ± 0.1 ^a^	0.9 ± 0.1 ^a^	1.0 ± 0.1 ^a^
Final OD_600_	2.0 ± 0.1 ^a^	1.4 ± 0.2 ^b^	2.0 ± 0.1 ^a^	1.1 ± 0.1 ^c^
Final biomass (g/L)	0.8 ± 0.1 ^a^	0.6 ± 0.1 ^b^	0.8 ± 0.1 ^a^	0.4 ± 0.1 ^c^
q_glucose_ (mmol/g/h)	77 ± 7 ^a^	96 ± 4 ^b^	103 ± 5 ^b^	168 ± 5 ^c^
Y_acetate_ (mol/mol)	0.31 ± 0.01 ^a^	0.41 ± 0.01 ^b^	0.24 ± 0.01 ^c^	0.65 ± 0.04 ^d^
Y_lactate_ (mol/mol)	1.45 ± 0.06 ^a^	1.52 ± 0.01 ^a^	1.57 ± 0.06 ^a^	1.47 ± 0.42 ^a^
Y_formate_ (mol/mol)	0.39 ± 0.08 ^a^	0.47 ± 0.01 ^a^	0.31 ± 0.08 ^a^	0.44 ± 0.07 ^a^
Y_ethanol_ (mol/mol)	0.10 ± 0.00 ^a^	0.07 ± 0.02 ^a^	0.10 ± 0.00 ^a^	0.08 ± 0.02 ^a^
Y_succinate_ (mol/mol)	0.04 ± 0.00 ^a^	0.00 ± 0.00 ^b^	0.05 ± 0.00 ^a^	0.00 ± 0.00 ^b^

^abc^ Within a row, means ± SD without a common superscript significantly differ (Student’s *t test*, *p* ≤ 0.05).

**Table 2 antioxidants-10-00819-t002:** Cellular peptides for which significant fold-changes in Met(O) levels (ǀlog_2_ FCǀ ≥ 1.5, adjusted *p* value ≤ 0.05) were detected in cellular extracts from Δ*msrA*, Δ*msrAB* and Δ*msrA*Δ*msrAB* compared to WT at early exponential growth phase (EE), late exponential growth phase (LE) and stationary growth phase (S).

Molecular Function	Gene No	Protein Name	Description	Met(O)/ Total Met	Met(O) Peptide Detected by LC MS/MS ^a^	log_2_FC ^b^					
						EE		LE		S	
						Δ*msrA*	Δ*msrA*Δ*msrAB*	Δ*msrA*	Δ*msrA*Δ*msrAB*	Δ*msrA*	Δ*msrA*Δ*msrAB*
**Amino acid metabolism**	BC1238	TrpA	Tryptophan synthase, alpha subunit	1/5	EVQMPFVL**M**TYLNPVLAFGK					4.2	
**ATP synthesis**	BC5308	AtpA	F0F1 ATP synthase, alpha subunit	1/10	I**M**QVPVGK			2.3			
**Chaperone proteins**	BC0295	GroEL	Chaperonin	5/20	SALQNAASVAAMFLTTEAVVADKPEPNAPAMPDMGG**M**G**M**GGMGGMM					3.3	
					NVTAGANP**M**GLR					2.0	−3.0
					SSIAQVAAISAADEEVGQLIAEA**M**ER	3.0		3.9	4.5	4.0	
					A**M**LEDIAILTGGEVITEELGR	4.5	5.0	4.3	4.6	4.7	4.3
**Degradative enzymes**	BC1991	TgC	Putative murein endopeptidase	1/9	YKQS**M**DGTMQDIKK			−3.3			
**Fermentation**	BC4365	AdhE	bifunctional acetaldehyde-CoA alcohol dehydrogenase	4/30	QL**M**NHDGVALVLATGGAGMVK				3.2		
					**M**IDTLVNNGQQALQALESFTQEEIDNIVHEMALAGVDQHMPLAK			2.6	3.0		
					LPLISELKEIY**M**K	2.8		3.2	4.3		
	BC4870	Ldh2	L-lactate dehydrogenase	1/7	GI**M**DSGFDGIFLIATNPVDILTYVTWK		3.2			4.8	3.7
**Glycolysis** **Gluconeogenesis**	BC4599	PykI	Pyruvate kinase	1/16	AASTDEMLDTAIQTGMDAGLIGLGDTVVITAGVPVAETGTTNL**M**K	2.7		3.4	4.0	3.8	3.4
	BC4898 ^c^	Pgi	Glucose-6-phosphate isomerase	2/9	FSVLTPVGLLPIAVSGLNIEE**M**MK			3.1			
					FSVLTPVGLLPIAVSGLNIEEM**M**K	2.9	3.5	3.9	4.8	4.3	4.4
	BC5135	Eno	Phosphopyruvate hydratase	5/11	QLPTPM**M**NIINGGSHADNNVDFQEFMILPVGAPTFK			2.6			
					VNQIGTLTETFEAIE**M**AKR	3.1		3.0			
					QLPTPMMNIINGGSHADNNVDFQEF**M**ILPVGAPTFK	2.5	3.4	3.3	3.9		
					A**M**IELDGTPNKGK	−2.6		−2.6			-2.4
					LGANAILGVS**M**AVAHAAADFVGLPLYR	3.5	3.7	3.3	3.6		2.9
					A**M**IELDGTPNK			−2.9			
	BC5140	GapA2	Glyceraldehyde-3-phosphate dehydrogenase	4/9	GM**M**TTIHSYTNDQQILDLPHK						3.8
					GILGYSEEPLVSIDYNGCTASSTIDALSTMVMEGN**M**VK			3.0			
					G**M**MTTIHSYTNDQQILDLPHKDLR			3.0			
					GILGYSEEPLVSIDYNGCTASSTIDALSTMVMEGN**M**VK	3.0		4.1	4.0		
					AAAEN**M**IPTSTGAAK	−2.3		−2.3			
**Protein export**	BC4410	YajC	Preprotein translocase	1/6	AVAQ**M**QSELAK	−3.2					
**Translation apparatus**	BC0129	Tuf	Elongation factor Tu	10/12	CD**M**VDDEELLELVE**M**EVR		3.9				
					CD**M**VDDEELLELVEMEVRDLLSEYGFPGDDIPVIK		3.9				
					ETDKPFL**M**PVEDVFSITGR			2.8		4.3	3.1
					IIELMAEVDAYIPTPERETDKPFL**M**PVEDVFSITGR			4.6	4.8		
					IIEL**M**AEVDAYIPTPERETDKPFLMPVEDVFSITGR	4.2	4.2	4.4	5.1		
					NMITGAAQ**M**DGGILVVSAADGPMPQTR	3.4	3.9				
					QVGVPYIVVFLNKCDMVDDEELLELVE**M**EVR		3.5				
					TTDVTGIIQLPEGTE**M**VMPGDNIEMTIELIAPIAIEEGTK	3.1		3.6	3.9	4.5	4.3
					TTDVTGIIQLPEGTE**M**V**M**PGDNIEMTIELIAPIAIEEGTK	3.1		3.1	3.4	3.9	3.4
					TTDVTGIIQLPEGTE**M**VMPGDNIE**M**TIELIAPIAIEEGTK			4.0	4.3	3.7	3.6
					TTDVTGIIQLPEGTEMV**M**PGDNIEMTIELIAPIAIEEGTK					3.2	3.3
					TTDVTGIIQLPEGTEMVMPGDNIE**M**TIELIAPIAIEEGTK						2.9
					VGDVVEIIGLAEENASTTVTGVE**M**FR					5.1	4.0
					VGDVVEIIGLAEENASTTVTGVE**M**FRK			6.1	6.1	4.4	
	BC5471	RplI	50S ribosomal protein L9	1/4	QGLAAEATNSS**M**K			−5.0	−4.6	−4.7	−5.6
	BC0155	Rpm	50S ribosomal protein L36	1/2	V**M**VICENPK			−2.9			
**Uncategorized**	BC4182 ^c^	Gls24	Unknown	1/6	VEIAPEVIEVIAGIAAAEVEGVAA**M**R	2.7		3.2		3.9	3.2

^a^ Met(O) residues that showed significant level changes are indicated in bold. ^b^ Only significant log_2_FC changes are reported. ^c^ Shaded lines show cellular proteins for which significant abundance changes were detected: the two highlighted proteins showed increased abundance in the *msr* mutant (Appendix A).

**Table 3 antioxidants-10-00819-t003:** Extracellular peptides for which significant fold-changes in Met(O) level (ǀlog2 FCǀ ≥ 1.5, adjusted p value ≤ 0.05) were detected in samples from cultures of ΔmsrA, ΔmsrAB and ΔmsrAΔmsrAB compared to WT at early exponential growth phase (EE), late exponential growth phase (LE) and stationary growth phase (S).

Molecular Function	Gene No	Protein Name	Description	Met(O)/ Total Met	Met(O) Peptide Detected by LC MS/MS^a^	log_2_FC^b^								
						EE			LE			S		
						Δ*msrA*	Δ*msrAB*	Δ*msrA*Δ*msrAB*	Δ*msrA*	Δ*msrAB*	Δ*msrA*Δ*msrAB*	Δ*msrA*	Δ*msrAB*	Δ*msrA*Δ*msrAB*
**Cell surface biogenesis**	BC5196	CwlD	N-acetylmuramoylL-alanine amidase	3/8	**M**DTVVTS**M**TSTEGQLKELEK	−1.1	−1.1	−1.0						
					ILESDEDI**M**K	−0.4	−1.5	−0.7						
**Chaperone**	BC0295	GroEL	Chaperonin	1/20	A**M**LEDIAILTGGEVITEELGR	0.8	1.5	0.2						
**Degradative enzymes**	BC0556	ColG	Collagenase	1/6	GLEVVTQA**M**H**M**YPR	1.1	1.1	1.3						
					**M**KGQAIYDIMQGIDYDIQSYLTEAR	0.0	0.0	0.0	9.4	1.2	9.9			
	BC0670	PlcB	Phospholipase C	1/6	AEVTP**M**TGKR	2.3	−0.1	2.4	9.5	2.6	10.0			
	BC1193	PepF1	Oligoendopeptidase F	1/11	ALGLDELH**M**YDLYTPLVPEVK							23.0	15.0	18.9
	BC1991	TgC	Murein endopeptidase	1/9	NI**M**DQLYGEFNKIVDADEYVK				9.4	−0.4	8.7			
					NI**M**DQLYGEFNKIVDADEYVKYNVASTR	−1.0	−0.9	−1.0						
	BC2735	NprP2	Bacillolysin	2/10	GIGEDK**M**FDIFYYANTDELNMTSNFK				6.5	1.0	6.6			
					GIGEDKMFDIFYYANTDELN**M**TSNFK				7.5	1.7	7.5			
	BC5351	NprB	Bacillolysin	1/6	N**M**SDIYDYFKK	−1.5	−1.2	−0.4						
					GNGIYIYNANYADSLGGYSQAGYPGTLISSSTPNFADKEAAGA**M**K	2.5	0.3	2.6	7.7	1.5	8.2			
**Flagella**	BC1657 ^c^	FlaA	Flagellin	2/15	ILNEAGIS**M**LSQANQTPQMVSK	−3.1	0.0	−3.1						
					VQLSDASGDT**M**TIDSLNAK				−6.0	0.1	−8.9			
	BC1658 ^c^	FlaB	Flagellin	1/15	TNFNGNSFLDTTATPPGKDIEIQLSDASGDT**M**TLK	−1.7	−0.7	−2.4						
	BC1659 ^c^	FlaC	Flagellin	1/15	LDHNLNNVTSQATNMAAAASQIEDAD**M**AKEMSEMTK	−2.6	−0.9	−2.8						
**Glycolysis** **Gluconeogenesis**	BC4898	Pgi	Glucose-6-phosphate isomerase	1/9	FSVLTPVGLLPIAVSGLNIEE**MM**K				7.2	3.3	7.2			
	BC5135	Eno	Phosphopyruvate hydratase	2/11	LGANAILGVS**M**AVAHAAADFVGLPLYR				9.5	2.2	9.6			
					VNQIGTLTETFEAIE**M**AK	1.8	1.6	1.4						
**Exotoxins**	BC3101	HblB’	Hemolysin BL, component B’	2/15	GLD**M**VKIPFIPTLIAGGI**M**IGDAR	1.5	0.9							
					IPFIPTLIAGGI**M**IGDAR				7.9	0.7	7.9			
	BC3102	HblB	Hemolysin BL, component B	4/8	S**M**NAYSY**M**LIKNPDVNFEGITINGYVDLPGR				6.3	0.7	6.7			
					**M**KETLQK	0.3	−1.0	0.4						
					QLLDTLNGIVEYDTTFDNYYET**M**VEAINTGDGETLKEGITDLR				8.4	2.2	7.7			
					QLLDTLNGIVEYDTTFDNYYET**M**VEAINTGDGETLK				7.9	0.9	7.4			
	BC3103	HblL1	Hemolysin BL, component L1	3/8	**M**LQDFKGK				8.2	3.7	8.2			
					IGELS**M**KADR	1.7	−0.3	2.1						
					QWNT**M**GANYTDLLDNIDS**M**EDHKFSLIPDDLK	1.9	0.8	1.6	8.0	1.5	7.4			
	BC3523	HlyII	Hemolysin II	1/7	DSFNTFYGNQLF**M**K				0.9	−5.4	0.9			
	BC1809	NheA	Non hemolytic enterotoxin, A	1/9	VLNNN**M**IQIQTNVEEGTYTDSSLLQK				7.6	−0.3	8.2			
	BC5101	HlyI	Cereolysin	2/7	**M**TLDHYGAYVAQFDVSWDEFTFDQK	0.8	1.0	0.4						
					KV**M**VAAYK				6.3		6.3			
**Toxin-like**	BC5239	EntA	Enterotoxin, cell wall binding	1/5	VLTAMGHDLTANPN**M**K	−1.4	−1.4	−0.6						
**Uncategorized**	BC5027 ^c^		ErfK/srfK precursor	1/4	**M**YNNDIHWLFER	−1.6	−0.9	−1.0						

^a^ Met(O) residues that showed significant level changes are indicated in bold. ^b^ Only significant log_2_FC changes are reported. ^c^ Shaded lines show exoproteins for which significant abundance changes were detected: the four highlighted proteins showed decreased abundance in the *msr* mutant (Appendix A).

## Data Availability

The data presented in this study are openly available in the PRIDE partner repository under dataset identifiers PXD024888 for the cellular proteome of *B. cereus* ATCC 14579, PXD024927 for the cellular proteome of the Δ*msrAB* mutant, PXD024850 for the cellular proteome of Δ*msrA* mutant, PXD024849 for the cellular proteome of Δ*msrAB*Δ*msrA* mutant, PXD024702 for the exoproteome of *B. cereus* ATCC 14579, PXD024714 for the exoproteome of Δ*msrAB* mutant, PXD024847 for the exoproteome of Δ*msrA* mutant and PXD024848 for the exoproteome of the Δ*msrAB*Δ*msrA* mutant.

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
