# Peer review of "Methionine Sulfoxide Reductases Contribute to Anaerobic Fermentative Metabolism in Bacillus cereus"

_antioxidants, 2021, doi:10.3390/antiox10050819_

Round 1
Reviewer 1 Report
MDPI Antioxidants. B.cereus MsrA paper April 21
This manuscript describes the characterization of anaerobic, fermentative growth of Bacillus cereus wt and strains lacking either one or two of the peptide methionine sulfoxide reductases encoded by the B. cereus genome.
The authors indicate that the focus of the investigation is on MsrA, ad MsrAB has been previously characterized. The results demonstrate a role for MsrA in motility and repair of a variety of target proteins that are identified using MS/MS techniques.
This is a well-written and presented manuscript, however, there are a few issues that require attention.
In particular, while it is clearly stated that MsrA is located in the cytoplasm, the location of MsrAB is not clearly explained but is crucial to the interpretations. In some bacterial species this enzyme is located extracellularly, and in others can be either an intracellular enzyme or located in both areas.
A concern is Figure 3 and its interpretation – while there clearly appear to be higher levels of met-O in the delta msrAB strains and the double mutant, this graph is interpreted as indicating the MsrA has a more important role in protein repair because its loss leads to higher metO formation levels. This needs to be urgently clarified.
A model showing the cellular locations of both MsrA and MsrAB and the proposed effects of each enzyme during anaerobic growth would be very helpful.
Specific comments:
Introduction:
L48/49 – please reword, the statement that DorA ‘ only reduces Met_O’ is misleading. Met-O reduction was demonstrated in that paper ( from memory for protein-bound residues), but the demonstrated role of this enzyme in Rhodobacter species is in supporting anaerobic respiration with DMSO.
L51 Please explain the concepts I ‘ cyclic msr-dependent methionine oxidation/reduction – it is not self-evident.
L 55 – please add a reference after …’protein activation or inactivation.’
L 63 please replace hypothesized with proposed
L 66- what is the cellular location of MsrAB
Methods:
L 106-110 – insufficient detail for qPCR. Please add information on the assay chemistry used, and the normalization method used. How was RNA isolated from samples?
L165 – unclear ‘ the data were missing not at random’? meaning?
L166 – what type of analysis gave rise to this adjusted p-value? Please provide detail of statistical evaluation.
L 197 – details of the statistical analyses are insufficient – please specify which tests were used for what type of data, please also add details of the method used to the figure legends when relevant data are shown.
Results:
L205 – please reword. Analysis of the msrAb gene and the encoded MsrAb proteins has been described…etc etc
L208 – what is the % identity between the two msrA domains?
L220 ff – it is not clear how the gene expression data were normalized ( please see above, ‘relative level’ – how are units defined?). Also not clear is how many samples were analyzed an averaged – average of two technical replicates for each biological replicate performed in triplicate – please reword to clarify. Would this mean that in case of 3 biol replicates being used, 2x3=6 mRNA values were averaged?
Please add the type of statistical analysis used to analyze the data.
In the figure or legend add OD of culture at sampling time to allow matching with Figure 2.
L254/255 – is that difference in final OD statistically significant/ likely to have biological relevance?
L255/256 – biomass for msrA mutant significantly higher than double mutant – the averages are 0.4 and 0.5 +-0.1 – what test was used to deduce statistical significance based on such a small difference? Please remove statement or change it to whatever it was meant to be. O can see differences in msrAB compared to the msrA and double mutants, for example.
L263 – statistical test used for p<0.05 – in fact, it would be good if the actual p-value obtained would be given rather than the threshold used.
L266 – this insight should be clearly linked to the analyses of proteins that may be repaired by msrA or MsrAB.
L275/276 – the statement that Fig 3 Panel A showed no differences between metO formation in the different strains contradicts the data shown in the figure. Please revise – this line of interpretation continues at other places in the manuscript and needs to be updated.
L 278 – fix spelling double mutant name
Fig 3 – Panel labels are missing. Figure legend – please add type of statistical test used, and actual p values obtained.
L319 – please define what criteria are used to designate a Met -O residue as a substrate of Msr or MsrA, etc. What is the evidence for the remaining proteins that they are not MsrA substrates?
How can a cytoplasmic enzyme repair/ maintain/ control the function of extracellular proteins? (this should be clearly discussed either here or at the latest in the discussion.
L328 – this statement is not what is seen in Figure 3 where only a loss of MsrAB led to higher levels of met-O oxidation in intracellular proteins, please clarify/ correct.
Table2 -Figure legend incomplete – logFC2 change related to what/ abundance or Met-O formation? If the former, please add Met-O formation. What is the meaning of blank cells in the logFC column? Are they equal to the peptide not being detected? Why is the red color used in the peptides to highlight methionines explained, but not the use of red and green.
The same applies to Table 3.
L344ff – please connect damaged protein profile to earlier results in metabolic endproducts ( either here or in discussion)
L348 – how are residues classified as MsrA- Met-O targets. Unclear.
L351- how can cytoplasmic enzymes ( or a cytoplasmic enzyme as location of MsrAb is not reported) convert/ repair extracellular substrates?
L355 – msrB – should that be msrAB?
L355 indicating that ‘they’ can be – please define what ‘they’ refers to the met-O residues?
The entire paragraph and discussion of extracellular Msr substrates should include some insights into how it is envisaged that the Msr proteins would access these substrate molecules. Figure 3 B shows only low levels of change in extracellular proteins Met-O formation.
L372 – unclear- how would the MsrA affect abundance levels of flagellin proteins? Please clarify. Presumably this is not related to changes in gene expression of flagellin levels?
Discussion
L394 – How would higher levels of oxidative damage arise earlier during growth under anaerobic conditions? Please add some mechanistic insights.
L399 – again, it is implied that MsrAb does not alter the met-O formation profile, but that contradicts the data shown in Figure 3A.
L416/417 – perturbation of fermentative metabolism – what is the evidence? This could be linked to the reduced growth yield in msrA mutant strains ( Table 1)
L 422 – replace ‘was’ with ‘were’ unchanged
L425 – unclear causal relationship – an enzyme reaction can lead to formation of both ATP and NAD or NADH, the fact that a reaction produces ATP is not an exclusion criterion for contributing to redox reactions.
L 428 – what is the reason for the increase in glucose consumption, especially in the double mutant where values were nearly doubled compared to the WT? Please add proposed mechanistic interpretations. Increased ATP demand might be due to an increased demand for ATP-consuming protein repair reactions involving molecular chaperones?
L436 – justify somewhere in this discussion the proposed increase in ATP demand in the mutant strains, as the reason for this proposal is not obvious.
L437FF – please indicate how MsrA is supposed to physically interact with extracellular substrates, be they exotoxins or flagella subunits. Please also explain how the function of MsrAB and MsrA overlap with regard to exotoxin production, which is the stated main function for MsrAB.
L447 – presence of MsrA and MsrAB in cells – this seems counterintuitive given that expression of the encoding genes changes with growth phase. The proteome samples were taken at what growth stage? What was the overall proteome coverage achieved for the samples used in this analysis?
L448 – ‘ no difference in their expression levels was observed’ – please clarify – is this relative to each other, because relative to the strain growth phase changes should have been observed ( Fig.1)
L452 – please explain how oxidative stress would chemically lead to a preferential formation of Met-S-O over Met-R-O. Include details of/ reference to the proposed mechanism for this.
L457ff – please clarify – is your proposal that the two Msr proteins ‘repair’ the exoproteins before they are exported/ folded/ functional? How would that explain the swimming/ motility phenotype? If that were the case, all exported flagella subunits should be in perfect working order.
Please include a model that summarizes your proposal regarding the function of the two enzymes.
References – formatting is inconsistent – journal names – use both abbreviation and full names
Titles- use both headline and sentence-style interchangeably.
Please amend.
Author Response
We appreciated the positive overall evaluation of our work and the constructive comments of reviewers that helped us to improve the presentation of our data.
we have answered to the reviewer's comments below
Reviewer 1:
In particular, while it is clearly stated that MsrA is located in the cytoplasm, the location of MsrAB is not clearly explained but is crucial to the interpretations. In some bacterial species this enzyme is located extracellularly, and in others can be either an intracellular enzyme or located in both areas.
Authors: Bioinformatic analysis predicted MsrA and MsrAB as cytoplasmic proteins in B. cereus. Unlike the surface-exposed pneumococcal MsrAB2, B. cereus MsrAB does not have a transmembrane domain at its N-terminus. Unlike the periplasmic MsrAB in Neisseria gonorrhoeae, B. cereus MsrAB has not a fused thioredoxin lipoprotein domain at its N-terminus. In S. pneumonia and N. gonorrhoae MrsAB production is coupled to the production of its redox partner. Such coproduction is needed to maintain the catalytic activity of MsrAB (via its cysteines) in an oxidizing environment such as the periplasm in gram negative and extracellular compartment in gram positive bacteria.
As reported both in this study and our previous study (Madeira et al, 2017), we have performed shotgun proteomics to get confirmation of MsrAB localization. We detected more than 2 peptides assigned to both MsrAB and MsrA in B. cereus cellular proteome under oxic conditions, and are thus confident of their identification and localization. No MsrAB and MsrA peptides was found in the exoproteome whatever the growth phases and the oxygenation conditions. These results indicate that MsrAB, like MsrA, is a cytoplasmic protein and we reported clearly this result in the revised version of the manuscript (lines 230-231) as followed. “The same analysis of an oxic cellular proteome detected 2 and 5 peptides for MsrA and MsrAB, respectively [25]. No peptide was detected in exoproteome, whatever the oxygenation conditions, indicating that both MsrA and MsrAB are cytoplasmic proteins”.
Reviewer 1:
A concern is Figure 3 and its interpretation – while there clearly appear to be higher levels of met-O in the delta msrAB strains and the double mutant, this graph is interpreted as indicating the MsrA has a more important role in protein repair because its loss leads to higher metO formation levels. This needs to be urgently clarified.
Authors : There were some inversions between the blue and green colors in Figure 3. We are sorry for this and we like to thank the reviewer for pointing out this mistake. We have corrected this inaccuracy. The corrected figure 3 shows unambiguously that MsrA has the most important role in regulating Met(O) level.
Reviewer 1:
A model showing the cellular locations of both MsrA and MsrAB and the proposed effects of each enzyme during anaerobic growth would be very helpful.
Authors: As recommended by the reviewer, we proposed a new figure (figure 4) showing the cellular locations of MsrA and MsrAB and their proposed effects on cellular metabolism.
Specific comments:
Introduction:
Reviewer 1:
L48/49 – please reword, the statement that DorA ‘ only reduces Met_O’ is misleading. Met-O reduction was demonstrated in that paper ( from memory for protein-bound residues), but the demonstrated role of this enzyme in Rhodobacter species is in supporting anaerobic respiration with DMSO.
Authors : As recommended by the reviewer, we modified the sentence as follows :” The fourth type of Msr is Rhodobacter sphaeroides DorA DMSO reductase, which reduces both free and protein-bound Met-S-O”
Reviewer 1:
L51 Please explain the concepts I ‘ cyclic msr-dependent methionine oxidation/reduction – it is not self-evident.
Authors: “Cyclic” refers to the fact that Msr proteins undergo cycles of oxidation and reduction. This formulation was proposed by a professional, native-English-speaking editor.
Reviewer 1:
L 55 – please add a reference after …’protein activation or inactivation.’
Authors: We added a new reference (Cui et al., 2012) as recommended by the reviewer.
Reviewer 1:
L 63 please replace hypothesized with proposed
Authors :We made the requested modification
Reviewer 1:
L 66- what is the cellular location of MsrAB
Authors : MsrAB is a cytoplasmic protein et we made it clear in the new version of the manuscript : “The first of these genes, msrAB, encodes a bifunctional cytoplasmic MsrAB enzyme, and is probably the result of gene fusion”
Methods:
L 106-110 – insufficient detail for qPCR. Please add information on the assay chemistry used, and the normalization method used. How was RNA isolated from samples?
Authors :
We used the method described in reference 26. We thought it was not necessary to repeat the whole description to avoid plagiarism. However, we added the requested details in the new version of the manuscript as follows :” Briefly, real-time RT-PCR was performed using the iScript™ One-Step RT-PCR kit with SYBR® Green following the manufacturer's protocol (Biorad), with 10 ng of total RNA as a template. Total RNA was extracted from B. cereus cells at EE, LE and S growth phases by using TRI Reagent RNA extraction solution as recommended by the manufacturer (Ambion). The Agilent 2100 Bioanalyzer system was used to characterize and quantify total RNA. The mRNA level changes for each gene were normalized to the RNA level for the ssu gene, encoding 16S RNA, and quantified by the 2−ΔΔCT method”.
Reviewer 1:
L165 – unclear ‘ the data were missing not at random’? meaning?
Authors: We agree with the reviewer: our previous sentence was not clear. This sentence is now deleted in the new version of the manuscript. We added a new sentence explaining how missing values (which are not randomly distributed across the samples) were imputed. The sentence is: “Missing values were imputed using random draws from a Gaussian distribution centered around a minimal value”.
Reviewer 1:
L166 – what type of analysis gave rise to this adjusted p-value? Please provide detail of statistical evaluation.
Authors: p-values were calculated using the Limma package, which is included in the DEP package. We added this detail in the new version of the manuscript:” Relative protein abundances were calculated based on spectral counts after correction and normalization by variance stabilizing (vsn), using the Limma package.”
Reviewer 1:
L 197 – details of the statistical analyses are insufficient – please specify which tests were used for what type of data, please also add details of the method used to the figure legends when relevant data are shown.
Authors: We provided the details requested by the reviewer in the new version of the manuscript, in the material and method section, figure legends and text when needed.
Results:
Reviewer 1:
L205 – please reword. Analysis of the msrAb gene and the encoded MsrAb proteins has been described…etc etc
Authors: We changed the sentence as follows:” The first gene, msrAB (BC 5436) has been described elsewhere and encodes a cytoplasmic MsrAB protein”.
Reviewer 1
L208 – what is the % identity between the two msrA domains?
Authors: The two domains share 64% sequence identity. We indicated this value in the new version of the manuscript.
Reviewer 1
L220 ff – it is not clear how the gene expression data were normalized ( please see above, ‘relative level’ – how are units defined?). Also not clear is how many samples were analyzed an averaged – average of two technical replicates for each biological replicate performed in triplicate – please reword to clarify. Would this mean that in case of 3 biol replicates being used, 2x3=6 mRNA values were averaged?
Please add the type of statistical analysis used to analyze the data.
In the figure or legend add OD of culture at sampling time to allow matching with Figure 2.
Authors: The data were normalized using the ssu gene as described in the material and method section in the new version of the manuscript. We also added this information in the legend of figure 1.
We analyzed 6 mRNA samples (3 biological replicates x 2 technical replicates). We reported this information in the material and method section and in the legend of the figure 1.
The type of statistical analysis used to analyze the data was added in the material and method section, and in the legend of figure 1.
In the legend of Figure 1, we indicated that the samples were harvested as indicated in Figure 2. This avoids repeating the same information within the manuscript (which is incorrect), while adding the nine OD values would inflate the legend size. However, we added in the legend of fig. 2 that samples were harvested as indicated in figure 2.
Reviewer 1
L254/255 – is that difference in final OD statistically significant/ likely to have biological relevance?
Authors: Statistical significance for OD and other parameters were added in Table 1 in the new version of the manuscript.
Reviewer 1
L255/256 – biomass for msrA mutant significantly higher than double mutant – the averages are 0.4 and 0.5 +-0.1 – what test was used to deduce statistical significance based on such a small difference? Please remove statement or change it to whatever it was meant to be. O can see differences in msrAB compared to the msrA and double mutants, for example.
Authors: The mean value for msrA mutant was 0.57. The rounded value is therefore 0.6 and not 0.5. We corrected this value in Table1. This value is significantly different to the value measured for the double mutant, and we indicated this significance in Table 1 in the new version of the manuscript
Reviewer 1
L263 – statistical test used for p<0.05 – in fact, it would be good if the actual p-value obtained would be given rather than the threshold used.
Authors: The significance of the differences is now shown in the revised Table 1. This should make the data easier to read and validate.
Reviewer 1
L266 – this insight should be clearly linked to the analyses of proteins that may be repaired by msrA or MsrAB.
Authors: This link is discussed in the discussion section.
Reviewer 1
L275/276 – the statement that Fig 3 Panel A showed no differences between metO formation in the different strains contradicts the data shown in the figure. Please revise – this line of interpretation continues at other places in the manuscript and needs to be updated.
Authors : We corrected figure 3, and used letters instead of asterisks to show significant differences between values. The new presentation of Figure 3 shows clearly that when each strain is taken individually, there is no change in the amount of Met (O) over time. On the other hand, for a given growth phase, the Met (O) content is not the same for all strains.
Reviewer 1
L 278 – fix spelling double mutant name
Authors : We corrected the typographical mistake in the double mutant name.
Reviewer 1
Fig 3 – Panel labels are missing. Figure legend – please add type of statistical test used, and actual p values obtained.
Authors : Panel labels (a and b) are indicated below the figure, as recommended by the journal. We added in the legend the information concerning the statistics
Reviewer 1
L319 – please define what criteria are used to designate a Met -O residue as a substrate of Msr or MsrA, etc. What is the evidence for the remaining proteins that they are not MsrA substrates?
Authors :A protein was considered as a Msr substrate if (i) its Met (O) level increases in the absence of Msr, and (ii) Met(O) level increase is not due to an increase of protein abundance. This was explained in the manuscript, lines 347-368 in the new version of the manuscript.
Our criteria identified some Msr substrates with high confidence, but certainly not all Msr substrates.
Reviewer 1
How can a cytoplasmic enzyme repair/ maintain/ control the function of extracellular proteins? (this should be clearly discussed either here or at the latest in the discussion.
Authors: A cytoplasmic protein can repair/ maintain/ control the function of extracellular proteins before its secretion. It was discussed in the discussion section, and explained in our previous study (Madeira et al., 2017).
Reviewer 1
L328 – this statement is not what is seen in Figure 3 where only a loss of MsrAB led to higher levels of met-O oxidation in intracellular proteins, please clarify/ correct.
Authors: Figure 3 considers the sum of Met (O). An absence of difference on the sum does not preclude individual differences, which are shown in Table 3. The text refers to Table 3 and from this point of view there is no correction to be made.
Reviewer 1
Table2 -Figure legend incomplete – logFC2 change related to what/ abundance or Met-O formation? If the former, please add Met-O formation. What is the meaning of blank cells in the logFC column? Are they equal to the peptide not being detected? Why is the red color used in the peptides to highlight methionines explained, but not the use of red and green.
The same applies to Table 3.
Authors: Fold-changes in Met(O) levels in mutant strains are related to WT as indicated in the tittles of table 2 and table 3. Only significant FC were reported. Blank cells indicate that the FC was no significant. We highlighted Met residues (bold characters) for which we detected Met(O) level changes.
We completed the legends of Tables 2 and 3 as recommended by the reviewer.
Reviewer 1
L344ff – please connect damaged protein profile to earlier results in metabolic endproducts ( either here or in discussion)
Authors:We connected damaged protein profile to earlier results in metabolic endproducts in discussion section.
Reviewer 1
L348 – how are residues classified as MsrA- Met-O targets. Unclear.
Authors: A Met(O) residue was considered as a MsrA target if its level increases significantly in the absence of the MsrA, while no change was observed in protein abundance.
Reviewer 1
L351- how can cytoplasmic enzymes ( or a cytoplasmic enzyme as location of MsrAb is not reported) convert/ repair extracellular substrates?
Authors: MsrAB is cytoplasmic, as now clearly indicated in the new version of the manuscript. It can reduce Met(O) of the extracellular proteins before they are secreted, even more so easily because the extracellular proteins acquire their 3D structure only after their export.
Reviewer 1
L355 – msrB – should that be msrAB?
Authors: We replaced mrsB by msrAB.
L355 indicating that ‘they’ can be – please define what ‘they’ refers to the met-O residues?
Authors: “they” refers to Met(O) residues and we specified it in the new version of the manuscript.
Reviewer 1
The entire paragraph and discussion of extracellular Msr substrates should include some insights into how it is envisaged that the Msr proteins would access these substrate molecules. Figure 3 B shows only low levels of change in extracellular proteins Met-O formation.
Authors: We discussed the role of Msr proteins in Met(O) level regulation of extracellular in our previous work (Madeira et al., 2017). We also discuss this point in the discussion section.
As already indicated in our responses here-above, figure 3 does not consider the proteins separately. An absence of global variation does not prevent individual variations.
Reviewer 1
L372 – unclear- how would the MsrA affect abundance levels of flagellin proteins? Please clarify. Presumably this is not related to changes in gene expression of flagellin levels?
Authors: An explanation could be that flagellin proteins could be over oxidized in the absence of MsrA, and then degraded, as reported in the discussion section. Another explanation could be that by decreasing flagellin synthesis B. cereus save energy to repair proteins in the absence of MrsA.
We did not examine the expression of flagellin genes.
Discussion
Reviewer 1
L394 – How would higher levels of oxidative damage arise earlier during growth under anaerobic conditions? Please add some mechanistic insights.
Authors: There is no data to explain why oxidative damage occurs earlier in anaerobiosis. It would therefore be too speculative to propose mechanisms.
Reviewer 1
L399 – again, it is implied that MsrAb does not alter the met-O formation profile, but that contradicts the data shown in Figure 3A.
Authors: Figure 3A was corrected, and the new figure 3A does not show difference between WT and msrAB mutant. In addition, no protein showed significant Met(O) level change in msrAB mutant compared to WT. Therefore, there is no contradiction between data and our statement.
Reviewer 1
L416/417 – perturbation of fermentative metabolism – what is the evidence? This could be linked to the reduced growth yield in msrA mutant strains (Table 1)
Authors: Carbon flow through fermentative pathways was modified in msrA mutant strains, indicating a perturbation of fermentative metabolism. Table 1 shows that there are significant changes in carbon flow through acetate, and succinate pathways. In the result section, we indicated “Taken together, these data suggest that MsrA mediates a much more significant regulatory effect on fermentative metabolism than MsrAB, and that together the two proteins may contribute to regulating carbon flow at the pyruvate node”.
Reviewer 1
L 422 – replace ‘was’ with ‘were’ unchanged
Authors: We replaced “was” by “were”.
Reviewer 1
L425 – unclear causal relationship – an enzyme reaction can lead to formation of both ATP and NAD or NADH, the fact that a reaction produces ATP is not an exclusion criterion for contributing to redox reactions.
Authors: Acetate pathway leads to formation of ATP, without oxidation of coenzymes. Reactions involved in this pathway are not oxidoreduction reactions. Acetate pathway is the sole fermentative pathway that produce ATP. Succinate pathways leads to oxidation of NADH, without production of ATP. In this case, there is no coupling between ATP and oxidation of coenzymes. However, we have changed the text to avoid any ambiguity.
Reviewer 1
L 428 – what is the reason for the increase in glucose consumption, especially in the double mutant where values were nearly doubled compared to the WT? Please add proposed mechanistic interpretations. Increased ATP demand might be due to an increased demand for ATP-consuming protein repair reactions involving molecular chaperones?
Authors: Increase in glucose consumption sustains higher glycolytic flow and then higher carbon flow through the acetate pathway, probably because a higher ATP demand is required to sustain mechanisms that have become less efficient due to the absence of Msr. We proposed such explanation in the discussion section.
Reviewer 1
L436 – justify somewhere in this discussion the proposed increase in ATP demand in the mutant strains, as the reason for this proposal is not obvious.
Authors: We proposed in line 457 of the new version of the manuscript an explanation for increased ATP demand.
Lactate and ethanol production were unchanged in the absence of MsrA, whereas succinate formation was decreased. However, due to the minor impact of the succinate pathway on NAD+ generation, this decrease probably has neutral effect on the cellular redox balance. The acetate pathway does not involve redox reactions and generates additional ATP outside of glycolysis, through the conversion of acetyl-CoA to acetate (Fig. 4). In the absence of MsrA, acetate secretion is increased, indicating a higher carbon flow through acetate pathway, and a concomitantly higher ATP production. To sustain high carbon flow through the acetate pathway, DmsrA and DmsrADmsrAB mutants increase carbon flow through glycolysis, as revealed by increased glucose consumption (Table 1). Finally, when MsrA is lacking, B. cereus adjusts its fermentative metabolism to maintain redox balance and promote ATP synthesis, probably to sustain - for as long as possible - repairing processes that have become less efficient due to the absence of Msr.
Reviewer 1
L437FF – please indicate how MsrA is supposed to physically interact with extracellular substrates, be they exotoxins or flagella subunits. Please also explain how the function of MsrAB and MsrA overlap with regard to exotoxin production, which is the stated main function for MsrAB.
Authors: Cytoplasmic Msr interact with exoproteins probably before their exportation, and we explained how the two Msr can interact in the last paragraph of the discussion section.
Reviewer 1
L447 – presence of MsrA and MsrAB in cells – this seems counterintuitive given that expression of the encoding genes changes with growth phase. The proteome samples were taken at what growth stage? What was the overall proteome coverage achieved for the samples used in this analysis?
Authors: We deleted the sentence that raised problems because it was out of the context.
Proteome samples were taken at EE, LE and S growth phases, and, whatever the growth phase, the overall proteome coverage was more than 50% thanks to our robust analytical procedure.
Reviewer 1
L448 – ‘ no difference in their expression levels was observed’ – please clarify – is this relative to each other, because relative to the strain growth phase changes should have been observed ( Fig.1)
Authors: This sentence was deleted in the new version of the manuscript because it was not appropriate and did not provide tangible elements for the mechanism of action of Msr.
Reviewer 1
L452 – please explain how oxidative stress would chemically lead to a preferential formation of Met-S-O over Met-R-O. Include details of/ reference to the proposed mechanism for this.
Authors: How oxidative stress leads to a preferential formation of Met S-O over Met-R-O is an interesting question. However, this is out of discussion context.
Reviewer 1
L457ff – please clarify – is your proposal that the two Msr proteins ‘repair’ the exoproteins before they are exported/ folded/ functional? How would that explain the swimming/ motility phenotype? If that were the case, all exported flagella subunits should be in perfect working order.
Authors: If the Msr proteins are lacking, they can’t repair proteins before their exportation. We showed that in the absence of MsrA, flagellin production and motility were decreased. Decrease of flagellin abundance can be due to overoxidation of their methionine and their subsequent degradation, but also to another regulation mechanism. Not all extracellular proteins are necessarily Msr substrates.
Reviewer 1
Please include a model that summarizes your proposal regarding the function of the two enzymes.
Authors: We included a new figure (figure 4) that summarizes our proposal.
Reviewer 1
References – formatting is inconsistent – journal names – use both abbreviation and full namesTitles- use both headline and sentence-style interchangeably.
Authors: We followed the instructions of MDPI, and used the MDPI endnote style.
Reviewer 2 Report
The aim of this work was to characterize the role of the antioxidant enzymes Msr in the facultative anaerobe Bacillus cereus. In particular, the authors examined the role of MsrA, having previously characterized the role of the bifunctional MsrAB enzyme in this pathogen, by generating both single and double Msr deleted mutants and investigating the effect of the deletion on B. cereus growth under fermentative anoxic conditions.
A number of issues should be addressed.
Line 95: to avoid confusion with the acronym indicating the synthesis phase of the cell cycle, the stationary growth phase should be indicated by a letter different than “S”. Maybe “St” ?
Line 207: "with a theoretical molecular weight of 20 475 Da." --> "with a theoretical molecular mass of 20 475 Da."
Line 208: "This protein shares an MsrA domain with the bifunctional MsrAB." The domain is not "shared", being MsrA and MsrAB two different polypeptides. The degree of homology and identity of MsrA with the MsrA domain of MsrAB must be given.
Lines 210-217. Following this part is difficult as it switches back and forth from aerobiosis to anaerobiosis. Fig 1S should be combined with Fig 1 in the main text to allow immediate comparison of Msr expression levels under the two growing conditions.
Lines 226-231. The most critical part of the present manuscript is the quantitative conclusions about levels of Msr expression. Accurate protein quantitation by proteomics is difficult: in this respect, the weakness of the present manuscript is in the lack of a more direct quantification of Msr expression, by enzymatic assay or at least Western blotting. After such a sophisticate proteomics approach what could only be concluded is that “… both Msr proteins are expressed at low abundance, and that their levels are reduced when cells are grown in anaerobic conditions compared to aerobic growth conditions.”
Lines 254-266: the significance of statistical analysis, where present, must be fully reported for data in Table 1, not only the result of statistical comparison for acetate production.
Similarly, as there may seem to be a significant difference in the final biomass among the various strains (Fig 2), the full statistical comparison should be provided.
Lines 276-280: description of the difference in MetO levels among strains is inconsistent with what presented in the graph, probably due to a mistake in labelling the histograms.
Lines 284-285: “Taken together, these data show that MsrA is a major contributor to the reduction of cellular protein-bound Met(O) under anaerobiosis, whereas MsrAB appears to play a minor role in this process.” Again, unless colors have been exchanged by mistake in Fig. 3, these statements are in conflict with the graph.
From Line 297: Identification of Msr substrate proteins. There are several issues in this part. First, it is impossible for the reader to draw the same conclusions as the authors did from the raw data. Table 2 is supposed to give the peptides whose Met(O) content changed more than 1.5 times in any given mutant strain with respect to WT.
Take for instance the putative cytoplasmic protein Gls24 (the peptide VEIAPEVIEVIAGIAAAEVEGVAAMR, because there are other two peptides on which the authors do not comment). What are the numbers in Table 1S under the column “Met(O)”? in WT all the values are zero. In MsrA mutant, LE phase, the average of the three samples gives 17.7. But what does this number mean? Is it the quantity of peptide detected with the single Met in the peptide oxidised to Met(O)?. But how is then the “fold change” calculated? As “fold change” is the ratio between the final quantity and the initial quantity, it will always be zero if the starting value is zero!
Then why give the fold change as the logarithm base 2? The tables must report plain “fold change”, not the logarithm, after explanation of how it is calculated.
But no matter how the data have been elaborated, the only conclusion that is drawn is that only a couple of proteins among several dozens show an increase in their Met(O) content in the mutants. Some actually show a decrease. Of the two that display the increase, one is an unknown cytoplasmic protein. The other is sloppily defined as glucose 6 phosphate dehydrogenase in the text (Line 313) but it is actually phosphoglucose isomerase (sic)!
Line 324 “a higher Met(O) content was detected for 32 Met residues compared to the levels present in WT.” please point to these 32 Met residues and their peptides (are they shown in Table 2?). Explain how was the increase in Met(O) content evaluated.
Line 325: “The abundance levels for the corresponding 13 proteins was unchanged (Table S4).” Make a separate list of the 13 proteins with a clear indication of the increase of Met oxidation that the authors can extrapolate from the raw data. As it stands, the presentation of the results is incomprehensible.
Line 315-318: How is it explained that all flagella proteins end up being more reduced in the mutants than in the WT (Table 3)? If this is because flagella proteins are also less expressed in the mutants, then this means that data in Table 2 and 3 are even more difficult to understand. In fact, it was assumed that the “fold change” in oxidation/reduction of Met in the proteins listed in the Tables was already normalized over protein abundance. Please explain.
Lack of oxidised Met in WT would point at an important role of “B” type of Msr, as MsrA alone would not be able to reduce the R diastereomer. So the role of MsrAB should not be underestimated, as it would appear form the conclusions of the paper. In addition, it appears that the presence of only MsrA and MsrAB in B. cereus was established on the basis of in silico studies. Can the authors exclude the existence of other Msr activities? Enzymatic assay of double mutant crude extracts could give some hints.
Minor
Line 25: Better to qualify from the beginning Bacillus cereus as a Gram-positive bacterium.
Line 32 "due to the reduced oxidative fermentation taking place." Better to avoid the ter "reduced" for possible confusion with redox cencept. "due to a decrease in oxidative phosphorylation"
Line 42-45: it is suggested that another reference is added about Msr discovery and their classification: Achilli C, et al., The discovery of methionine sulfoxide reductase enzymes: an historical account and future perspectives. Biofactors 2015;41:135-152. DOI: 10.1002/biof.1214.
Line 48: "The fourth type of Msr is Rhodobacter sphaeroides DorA DMSO reductase, which only reduces Met-S-O [12]". Why "only"? More explicitly: "which reduces both free and protein-bound Met-S-O"
Figure 3: “a” and “b” letters in Figure 3 should be uppercase
Author Response
We provided a point-by-point response to the reviewers's comments as follows :
Reviewer 2:Line 95: to avoid confusion with the acronym indicating the synthesis phase of the cell cycle, the stationary growth phase should be indicated by a letter different than “S”. Maybe “St” ?
Authors: We used the letter” S” for stationary growth phase in several previous studies, and we prefer to keep this abbreviation to avoid any confusion with our previous works. We believe there will be no confusion with the synthesis phase of the cell cycle, at least for microbiologists.
Reviewer 2:
Line 207: "with a theoretical molecular weight of 20 475 Da." --> "with a theoretical molecular mass of 20 475 Da."
Authors: We replaced “weight” by “mass”.
Reviewer 2:
Line 208: "This protein shares an MsrA domain with the bifunctional MsrAB." The domain is not "shared", being MsrA and MsrAB two different polypeptides. The degree of homology and identity of MsrA with the MsrA domain of MsrAB must be given.
Authors: We rewrote the sentence, as recommended by the reviewer and indicated the 64% of identity between the MsrA domains of MsrA and MsrAB in the new version of the manuscript.
Reviewer 2:
Lines 210-217. Following this part is difficult as it switches back and forth from aerobiosis to anaerobiosis. Fig 1S should be combined with Fig 1 in the main text to allow immediate comparison of Msr expression levels under the two growing conditions.
Authors: We appreciate the suggestion, but some of the results presented in Fig S1 were previously published. We therefore cannot put them back in the main text.
Reviewer 2:
Lines 226-231. The most critical part of the present manuscript is the quantitative conclusions about levels of Msr expression. Accurate protein quantitation by proteomics is difficult: in this respect, the weakness of the present manuscript is in the lack of a more direct quantification of Msr expression, by enzymatic assay or at least Western blotting. After such a sophisticate proteomics approach what could only be concluded is that “… both Msr proteins are expressed at low abundance, and that their levels are reduced when cells are grown in anaerobic conditions compared to aerobic growth conditions.”
Authors: We quantified msr expression by qRT-PCR. Quantify Msr production by enzymatic assay under anaerobic conditions is extremely difficult. Western blot analysis requires to have antibodies against these proteins. There is no longer a need to validate proteomic studies with biochemical studies, the methodology being mature and widely accepted (see the editorial doi: 10.1074/mcp.E113.031658). Anyway, regardless of the amount of Msrs, our results clearly demonstrate their roles in anaerobic fermentative metabolism of B. cereus.
Reviewer 2:
Lines 254-266: the significance of statistical analysis, where present, must be fully reported for data in Table 1, not only the result of statistical comparison for acetate production.
Similarly, as there may seem to be a significant difference in the final biomass among the various strains (Fig 2), the full statistical comparison should be provided.
Authors: As recommended by both reviewers, we modified Table I and Fig. 2 to show the significance of the data.
Reviewer 2:
Lines 276-280: description of the difference in MetO levels among strains is inconsistent with what presented in the graph, probably due to a mistake in labelling the histograms.
Authors: there was indeed a mistake in labelling the histograms. We corrected these mistakes, and there is no longer inconsistency between description of the data and fig.3.
Reviewer 2:
Lines 284-285: “Taken together, these data show that MsrA is a major contributor to the reduction of cellular protein-bound Met(O) under anaerobiosis, whereas MsrAB appears to play a minor role in this process.” Again, unless colors have been exchanged by mistake in Fig. 3, these statements are in conflict with the graph.
Authors: as indicated in our answers here-above, we corrected figure 3, and there is no longer inconsistency between description of the data and figure 3
Reviewer 2:
From Line 297: Identification of Msr substrate proteins. There are several issues in this part. First, it is impossible for the reader to draw the same conclusions as the authors did from the raw data. Table 2 is supposed to give the peptides whose Met(O) content changed more than 1.5 times in any given mutant strain with respect to WT.
Authors: Table 2 reports the values of log2 FC, and not the values of FC. The Met(O) content is thus significantly changes when log2 FC>0.58.
Reviewer 2:
Take for instance the putative cytoplasmic protein Gls24 (the peptide VEIAPEVIEVIAGIAAAEVEGVAAMR, because there are other two peptides on which the authors do not comment). What are the numbers in Table 1S under the column “Met(O)”? in WT all the values are zero. In MsrA mutant, LE phase, the average of the three samples gives 17.7. But what does this number mean? Is it the quantity of peptide detected with the single Met in the peptide oxidised to Met(O)?. But how is then the “fold change” calculated? As “fold change” is the ratio between the final quantity and the initial quantity, it will always be zero if the starting value is zero!
Authors : Only the peptides carrying Met (O) whose rate changes significantly are reported in Table 2. Table S1 reports the number of peptides containing oxidized methionine in the cellular proteome. Fold-changes were determined using the DEP package in R, as reported in the material and method. This R package allows to transform the row data, and then to calculate the fold-changes. Details concerning the DEP package was described at https://www.bioconductor.org . In brief, to be analyzed, row data must be normalized and missing values (i.e values =0) must be filtered and imputed. Missing values are usually representative of the detection limit of the instrumentation. A missing value is a problem in a relative comparison experiment, even if a peptide is quantified in one of the two conditions. It is not possible to calculate a ratio if one of the two values is missing. Imputation of missing values is therefore performed by replacing them with estimated ones, usually representing detection limit. There are multiple ways to perform data imputation, and we used the MNAR method proposed in DEP. Then, fold changes can be estimated even if a peptide is not detected in one of the two strains. All proteomics bioinformatic tools used such transformation of raw data. This is well explained in the article of Aguilan et al (https://doi.org/10.1039/D0MO00087F)
Reviewer 2:
Then why give the fold change as the logarithm base 2? The tables must report plain “fold change”, not the logarithm, after explanation of how it is calculated.
Authors: Results obtained from DEP analysis are given as the logarithm base 2, and presenting the results in this form is fairly standard. In addition, if someone wanted to redo the calculations, they could directly compare their results to those shown in the article.
Reviewer 2:
But no matter how the data have been elaborated, the only conclusion that is drawn is that only a couple of proteins among several dozens show an increase in their Met(O) content in the mutants. Some actually show a decrease. Of the two that display the increase, one is an unknown cytoplasmic protein. The other is sloppily defined as glucose 6 phosphate dehydrogenase in the text (Line 313) but it is actually phosphoglucose isomerase (sic)!
Authors : We have chosen to use strong statistical criteria, explaining why we found only 15 cellular and 21 extracellular proteins modified at the Met(O) level. Consequently, the significance of the results is very high.
We agree with the reviewer, there is an inconsistence between text and Table 2. We replaced in the text “glucose 6 phosphate dehydrogenase” by “glucose 6 phosphate isomerase” as indicated in the Table 2.
Reviewer 2:
Line 324 “a higher Met(O) content was detected for 32 Met residues compared to the levels present in WT.” please point to these 32 Met residues and their peptides (are they shown in Table 2?). Explain how was the increase in Met(O) content evaluated.
Authors: The Met residues are indicated in bold in column 6 of Tables 2 and 3 in the new version of the manuscript. We added foot-notes in Table 2, to facilitate Table reading. Increase of Met(O) level was evaluated using the DEP package as described in the material and method section.
Reviewer 2:
Line 325: “The abundance levels for the corresponding 13 proteins was unchanged (Table S4).” Make a separate list of the 13 proteins with a clear indication of the increase of Met oxidation that the authors can extrapolate from the raw data. As it stands, the presentation of the results is incomprehensible.
Authors: Table 2 lists the 15 proteins, with significant Met(O) level changes. Among them, 13 proteins did no shown abundance level changes, and 2 have shown abundance levels changes, as reported in Table S4. These two proteins are in grey in Table 2. We added a footnote in Table 2, explaining why these two proteins are in grey, while the 13 other proteins are in white.
Reviewer 2:
Line 315-318: How is it explained that all flagella proteins end up being more reduced in the mutants than in the WT (Table 3)? If this is because flagella proteins are also less expressed in the mutants, then this means that data in Table 2 and 3 are even more difficult to understand. In fact, it was assumed that the “fold change” in oxidation/reduction of Met in the proteins listed in the Tables was already normalized over protein abundance. Please explain.
Authors: We indicated in the text “Both the Met(O) level and overall abundance of four extracellular proteins was decreased in mutant strains compared to WT (Table 3). Among these proteins, three (flagellins FlaA, FlaB and FlaC) are components of the flagellar apparatus”. This suggests that the decrease Met(O) level of flagella protein is probably due to the decrease of their abundance. This point is discussed in the discussion section. Like in Table 2, we indicated in Table 3, proteins for which both Met(O) level and abundance level were significantly changes. These proteins are indicated in grey, and among these proteins are the flagellin proteins.
Reviewer 2:
Lack of oxidised Met in WT would point at an important role of “B” type of Msr, as MsrA alone would not be able to reduce the R diastereomer. So the role of MsrAB should not be underestimated, as it would appear form the conclusions of the paper. In addition, it appears that the presence of only MsrA and MsrAB in B. cereus was established on the basis of in silico studies. Can the authors exclude the existence of other Msr activities? Enzymatic assay of double mutant crude extracts could give some hints.
Authors: We did not underestimate the role of MsrAB. We only indicated that the role of MsrAB is less important in anaerobic fermentative metabolism than the role of MsrA. However, both the two proteins are important in regulating the Met(O) level of some virulence factors. The presence of MsrA and MsrAB was not established on the sole basis of in silico studies. We detected the mRNA coding for these two proteins, and we detected these two proteins in cellular proteome of B. cereus, using MS. We cannot exclude the existence of other Msr activities. However, the effects that we observed are only due to the absence of MsrA or/and MsrAB since we deleted the gene coding for these Msr, and showed that complementation of mutant strains restore the wild-type phenotype.
Minor
Reviewer 2:
Line 25: Better to qualify from the beginning Bacillus cereus as a Gram-positive bacterium.
Authors : We added this qualification as recommended by the reviewer.
Reviewer 2:
Line 32 "due to the reduced oxidative fermentation taking place." Better to avoid the ter "reduced" for possible confusion with redox cencept. "due to a decrease in oxidative phosphorylation"
Authors: We made the correction.
Reviewer 2:
Line 42-45: it is suggested that another reference is added about Msr discovery and their classification: Achilli C, et al., The discovery of methionine sulfoxide reductase enzymes: an historical account and future perspectives. Biofactors 2015;41:135-152. DOI: 10.1002/biof.1214.
Authors: We added this reference.
Reviewer 2:
Line 48: "The fourth type of Msr is Rhodobacter sphaeroides DorA DMSO reductase, which only reduces Met-S-O [12]". Why "only"? More explicitly: "which reduces both free and protein-bound Met-S-O".
Authors: We agree with the reviewer: the word “only” is not explicit. We corrected the sentence as recommended.
Figure 3: “a” and “b” letters in Figure 3 should be uppercase
Authors: we followed the journal's instructions which require the letters to be in lowercase.
Round 2
Reviewer 1 Report
This paper describes the roles of Msr-type peptide methionine sulfoxide reductases, MsrA and MsrAB, during anaerobic growth of Bacillus cereus.
The focus of the work is on MsrA, with the previously studied MsrAB being found to play a lesser role under these growth conditions.
The revised version of the manuscript has improved on several issues, but there are several issues that were previously raised and either not or partially addressed. A detailed list of comments that need to be addressed is set out below.
A major issue is the identification of large numbers of extracellular substrate proteins for MsrA which is, according to analyses also presented, an exclusively intracellular, cytoplasmic protein. This is an unusual aspect of the analyses, and I guess in most cases an analysis of extracellular substrate proteins for a cytoplasmic enzyme may not even be undertaken. However, as the data are presented it would be great if more details could be presented.
The proposed mechanism is that the proteins are oxidatively damaged following synthesis in the reducing environment of the cytoplasm, are repaired and then exported. Other than that Msr proteins preferentially re-reduce MetO in unfolded proteins (which can occur extra- and intracellularly following oxidative damage) and that the exoproteins are unfolded prior to export, very little in the way of a mechanistic explanation of this process is given. Additionally – the levels of extracellular oxidized proteins were reduced in the double mutant strain (all data points) and the msrA deletion strain (in LE, where expression of msrA is maximal and stationary phase). How do these overall results fit into the general analysis of metO formation and MsrA/ MSAB function? How would proteins be oxidatively damaged in the cytoplasmic environment directly after synthesis, and would that high level of damage then not affect all proteins produced in B. cereus?
Please amend the explanation, potentially by highlighting that this is a speculative, not a confirmed mechanism of action, or by providing data on precedents where repair of extracellular proteins has been shown to precede the export and functional formation of the extracellular protein.
Also – some of the changes made to address comments, such as the used of letters or combinations of letters to indicate different levels of statistical significance are not an improvement as it is a custom system of labelling, and no key is provided that would explain the meaning of the assignments of ‘a’, ‘b’, or ‘bc’ etc. to columns. This needs to be addressed.
Specific comments:
Introduction:
L42-47: Please clearly state that the MsrA & B proteins are peptide methionine sulfoxide reductases that use thiol chemistry to reduce MetO, while MsrP and DorA and both molybdenum enzymes, but of different families. If the sulfite oxidase enzyme family is mentioned for MsrP, why is the DMSO reductase enzyme family not mentioned for DorA.
L49-50: This statement ( all found enzymes have a primary role in repairing proteins) is simply incorrect. For DorA from various Rhodobacter species it has been clearly shown that this is an enzyme that reduces DMSO to support anaerobic respiration. The enzyme is produced in the presence of DMSO, which has been conclusively demonstrated already in the 1990s and early 2000s. The Km value for DMSO is at least 10-fold less than any of the Kms reported for oxidized Met or proteins, and DMSO occurs naturally in the environment where Rhodobacter species occur. The protein repair function has, to my knowledge, not been demonstrated in vivo for Rhodobacter species, despite the use of multiple substrate proteins.
Please reword this statement so that it reflects that fact that for DorA protein repair is not the major function, but may be a moonlighting function.
L67 – in view of the conclusions and the major role of MsrAB mutant strains in this study, it would be good to include more details about the earlier work on MsrAB, including under which conditions the role of the protein was studied, and whether intra- and extracellular proteins were identified. The previous work seems to have looked at aerobic growth conditions, but many of the proteins identified as substrates or differentially expressed are the same as those presented in this manuscript. This is an important point and should be discussed in the discussion.
Methods:
L106 ff ( qPCR) – comment: normalization against 16S is not recommended ( MIQUE guidelines), because of the significant difference in expression levels of 16S and genes encoding cellular proteins (this can be >10000x). As a result it seems interesting that the normalized expression is in whole numbers (Fig.1), as usually when 16S normalization is used the gene expression values are very small as a result of the large difference in gene expression levels.
Figure 1 and 3 – please provide a legend that details the meaning of the letters used to denote different levels of statistical significance, or use the more commonly used system where asterisks are used. Please also indicate what type of ANOVA was used in each case, 1-Way, 2-Way etc.
L261 – please add some quantitative statements – i.e. what is the fold change or percent change in final OD, or the change in biomass production observed?
Fig.3 – the order of the mutant strains in the graph does not match the order given in the Figure legend. In other analyses the mutants are presented in the order Wt, msrA, msrAB, double mutants (as set out in the figure legend).
L288 – would this observation have been expected? For mutations in cytoplasmic proteins, what are the expected changes in MetO formation for cytoplasmic and extracellular proteins? This is also important with a view to the later analyses, l 311 e.g..
L324 ff – please see above – it would be nice to explain why it would be expected that the cytoplasmic Msrs would reduce extracellular proteins.
L330 – why is a dot used rather than bold font for what seems to be a subheader?? Same in l 361?
L331 – indicates that oxidation levels of WT and msrAB mutant strain are shown in Table 2, but that s not the case. Please add te relevant columns.
Table 2 & 3 – the formatting of these tables could be significantly improved if the formatting used in the earlier paper on MsrAB were used. As is the fonts are too small to be legible, especially the fold changes and gene names. Please also explain whether the grey colour indicates increases or decreases with statistical significance.
L365/366 – what is the mechanisms (please see above), and how would MetO residue content be ‘regulated’ ( repaired?)
Discussion:
L406 – please add reference after ‘during the stationary phase.’ as hat data is not shown
L437/438 – what is meant by succinate pathway? If this is the part of the TCA cycle.
L438 ff – How is NADH balance maintained when acetate production increases?
L447/448 – could this be linked to the ATP synthase deficiency?
L455/456 – this needs a better explanation ( please see above), this interpretation is very speculative.
L467/468 – This is a major issue: It is not clear how it is chemically possible that one stereoisomer of methionine would form preferentially during chemically induced oxidative damage. I am not aware of a mechanism that would explain such a preferential formation. Please provide a chemically plausible explanation, details relating to the nature of oxidation on the specific substrate proteins that corroborate this suggestion, or remove this part of the explanation. It is, in as far as I am aware, an unresolved question why in different organisms, MsrA or MsrB proteins can have key roles for cell health, while the enzyme converting the other MetO stereoisomer is absent or has a less important role. My personal view on this is that it is likely related to the protein substrates that are being repaired, where on key proteins due to the particular environment of the methionine residue R- and S-MetO may form preferentially.
Author Response
We appreciated the comments of reviewers that helped us to improve the presentation of our data.
Our answers to the reviewer’s comments are given below.
Reviewer General comments
This paper describes the roles of Msr-type peptide methionine sulfoxide reductases, MsrA and MsrAB, during anaerobic growth of Bacillus cereus. The focus of the work is on MsrA, with the previously studied MsrAB being found to play a lesser role under these growth conditions. The revised version of the manuscript has improved on several issues, but there are several issues that were previously raised and either not or partially addressed. A detailed list of comments that need to be addressed is set out below.
A major issue is the identification of large numbers of extracellular substrate proteins for MsrA which is, according to analyses also presented, an exclusively intracellular, cytoplasmic protein. This is an unusual aspect of the analyses, and I guess in most cases an analysis of extracellular substrate proteins for a cytoplasmic enzyme may not even be undertaken. However, as the data are presented it would be great if more details could be presented.
The proposed mechanism is that the proteins are oxidatively damaged following synthesis in the reducing environment of the cytoplasm, are repaired and then exported. Other than that Msr proteins preferentially re-reduce MetO in unfolded proteins (which can occur extra- and intracellularly following oxidative damage) and that the exoproteins are unfolded prior to export, very little in the way of a mechanistic explanation of this process is given.
Authors: The objective of the work presented here was to determine the role of Msr in anaerobic fermentative metabolism. There is a strong link between fermentative metabolism and secretion of proteins, especially secretion of virulence factors in B. cereus. In addition, we showed in our previous paper that, under aerobiosis, MsrAB regulated Met(O) level of B. cereus exoproteome and we identified virulence factors as putative MsrAB substrates. For these reasons, we examined the role of Msrs both on cellular and extracellular proteins. We showed that absence of Msrs increased the Met(O) content of some virulence factors. Based on this result and the properties of Msr, we proposed that some virulence factors could be Msr substrates. This hypothesis is indeed speculative because we do not have any experimental data that prove it. In our opinion, going into the mechanistic details would be too speculative.
Reviewer :Additionally – the levels of extracellular oxidized proteins were reduced in the double mutant strain (all data points) and the msrA deletion strain (in LE, where expression of msrA is maximal and stationary phase). How do these overall results fit into the general analysis of metO formation and MsrA/ MSAB function? How would proteins be oxidatively damaged in the cytoplasmic environment directly after synthesis, and would that high level of damage then not affect all proteins produced in B. cereus?
Authors: We disagree with the reviewer comment. As illustrated in Figure 3, the levels of extracellular oxidized proteins were not significantly reduced in the mutant strains compared to WT (same letters indicated no statistical differences). Overall, the Met (O) content of the exoproteome is not changed in mutant strains compared to WT. However, some proteins have a higher Met (O) level in the absence of Msr, such as enterotoxins. The rate of synthesis of these proteins is very high at the end of growth, while the production of cellular proteins decreased. In addition, they could be more susceptible to Msr repair, compared to cellular proteins, because they are unfolded in the cytoplasm. Taken together these data suggest that some extracellular could be preferred target for MsrR before their exportation. However, this is only a hypothesis, and need to be prove.
Reviewer: Please amend the explanation, potentially by highlighting that this is a speculative, not a confirmed mechanism of action, or by providing data on precedents where repair of extracellular proteins has been shown to precede the export and functional formation of the extracellular protein.
Authors: We highlighted that our proposition is speculative in the discussion section, as recommended.
Reviewer 1:
Also – some of the changes made to address comments, such as the used of letters or combinations of letters to indicate different levels of statistical significance are not an improvement as it is a custom system of labelling, and no key is provided that would explain the meaning of the assignments of ‘a’, ‘b’, or ‘bc’ etc. to columns. This needs to be addressed.
Authors: There are two possibilities to show if two variables are significantly different or not: letters (a, b, c, ab…) and asterisks. If there is a lot of comparison, a figure is much more readable with letters. Using letters in a table is easier than asterisks. For these reasons, we have chosen to use letters in all cases. The convention is that for all variables with the same letter, the difference between the means is not statistically significant. If two variables have different letters, they are significantly different. Although it’s a convention, we indicated in the figure legends how to interpret these letters.
Reviewer -Specific comments:Introduction:
ReviewerL42-47: Please clearly state that the MsrA & B proteins are peptide methionine sulfoxide reductases that use thiol chemistry to reduce MetO, while MsrP and DorA and both molybdenum enzymes, but of different families. If the sulfite oxidase enzyme family is mentioned for MsrP, why is the DMSO reductase enzyme family not mentioned for DorA.
Authors: We followed reviewer recommendations. We changes the text as follows : Four types of Msr have been identified that reduce Met(O) residues to their functional form [13]. The main types are the thiol-oxidoreductases MsrA and MsrB, which react specifically with diastereomers displaying the S- and R-configurations at the sulfur atom, respectively [14,15]. A third type of Msr, periplasmic methionine sulfoxide reductase (MsrP), is present in most gram-negative bacteria and is a member of the sulfite oxidase family. MsrP is not a stereospecific protein-MetO reductase [16]. The fourth type of Msr is Rhodobacter sphaeroides DorA DMSO reductase, which reduces both free and protein-bound Met-S-O
Reviewer: L49-50: This statement ( all found enzymes have a primary role in repairing proteins) is simply incorrect. For DorA from various Rhodobacter species it has been clearly shown that this is an enzyme that reduces DMSO to support anaerobic respiration. The enzyme is produced in the presence of DMSO, which has been conclusively demonstrated already in the 1990s and early 2000s. The Km value for DMSO is at least 10-fold less than any of the Kms reported for oxidized Met or proteins, and DMSO occurs naturally in the environment where Rhodobacter species occur. The protein repair function has, to my knowledge, not been demonstrated in vivo for Rhodobacter species, despite the use of multiple substrate proteins.
Please reword this statement so that it reflects that fact that for DorA protein repair is not the major function, but may be a moonlighting function.
Authors: We modified the text as recommended by the reviewer, as follows “The primary role of the first three types of enzymes is to regulate the Met(O) level in proteins, they reduce Met(O) residues more efficiently in unfolded proteins than in folded proteins [17]. The role of DorA-like enzymes in protecting proteins against oxidation remains to be validated in vivo[13].
Reviewer: L67 – in view of the conclusions and the major role of MsrAB mutant strains in this study, it would be good to include more details about the earlier work on MsrAB, including under which conditions the role of the protein was studied, and whether intra- and extracellular proteins were identified. The previous work seems to have looked at aerobic growth conditions, but many of the proteins identified as substrates or differentially expressed are the same as those presented in this manuscript. This is an important point and should be discussed in the discussion.
Authors: We clearly demonstrate and indicate in our paper that this is MsrA and not MsrAB, that has the major role in anaerobic fermentative metabolism. This was first indicated in the abstract “MsrA therefore appears to play a major physiological role compared to MsrAB, placing methionine sulfoxides at the center of the B. cereus antioxidant system under anoxic fermentative conditions.” We thus think that include more details of our previous work on the role of MsrAB in aerobic conditions is not an important point for this study. We would also like to avoid any plagiarism.
Methods:
Reviewer: L106 ff ( qPCR) – comment: normalization against 16S is not recommended ( MIQUE guidelines), because of the significant difference in expression levels of 16S and genes encoding cellular proteins (this can be >10000x). As a result it seems interesting that the normalized expression is in whole numbers (Fig.1), as usually when 16S normalization is used the gene expression values are very small as a result of the large difference in gene expression levels.
Authors: We agree that there is a significant difference in expression levels of 16S gene and genes encoding cellular proteins. We used 16S gene as a reference gene because its expression did not change during B. cereus growth (we checked it in our experiments). In our study, we monitored the changes of msr gene expression during growth. We analyzed the data using the 2DDCT method, and early exponential growth phase (EE) was considered as the reference growth phase. Therefore, the mean fold-change at this growth phase is very close to one, and the values at LE and S growth phase are relative to the value observed at EE growth phase.
Reviewer: Figure 1 and 3 – please provide a legend that details the meaning of the letters used to denote different levels of statistical significance, or use the more commonly used system where asterisks are used. Please also indicate what type of ANOVA was used in each case, 1-Way, 2-Way etc.
Authors: We already indicated in fig 1 and 3 the meaning of the letters: “Different letters indicate a significant difference (p≤0.05) between strains at each growth phase”. We indicated in the new version of the manuscript that the type of ANOVA we used in each case (two-way ANOVA).
Reviewer: L261 – please add some quantitative statements – i.e. what is the fold change or percent change in final OD, or the change in biomass production observed?
Authors: we added the fold-decrease for msrA (~1.3 fold) and double mutant (2-fold) in the new version of the manuscript.
Reviewer: Fig.3 – the order of the mutant strains in the graph does not match the order given in the Figure legend. In other analyses the mutants are presented in the order Wt, msrA, msrAB, double mutants (as set out in the figure legend).
Authors: we changed the order of mutant strains in the graph as recommended by the reviewer.
Reviewer: L288 – would this observation have been expected? For mutations in cytoplasmic proteins, what are the expected changes in MetO formation for cytoplasmic and extracellular proteins? This is also important with a view to the later analyses, l 311 e.g..
Authors: In our previous study, we showed that, under aerobiosis, lack of MsrAB did not change the total Met(O) content of cytoplasmic proteins but changed the total Met(O)content of extracellular proteins. This result was surprising. In this study, which was performed under fermentative anaerobic conditions, lack of Msrs did not change the total Met(O) content of extracellular proteins. This result was neither expected nor predictable because (i) there are poor data on the role of Msr in fermentative conditions and, (ii) the exoproteome is not the same under anaerobiosis and aerobiosis. However, some extracellular proteins, such as toxins which are produced at high level both under aerobiosis and anaerobiosis showed Met(O) content changes in the absence of Msr in the two conditions.
Reviewer: L324 ff – please see above – it would be nice to explain why it would be expected that the cytoplasmic Msrs would reduce extracellular proteins.
Authors: We believe that the explanations should be given in the discussion section and not in the result section.
ReviewerL330 – why is a dot used rather than bold font for what seems to be a subheader?? Same in l 361?
Authors: We followed the Antioxidant template that proposes to use bullets for subsections. We changed the presentation in the new version of the manuscript, and used numbers as also proposed in the template.
Reviewer L331 – indicates that oxidation levels of WT and msrAB mutant strain are shown in Table 2, but that s not the case. Please add te relevant columns.
Authors: We preferred to remove the reference to table 2 rather than adding empty columns in table 2.
Reviewer: Table 2 & 3 – the formatting of these tables could be significantly improved if the formatting used in the earlier paper on MsrAB were used. As is the fonts are too small to be legible, especially the fold changes and gene names. Please also explain whether the grey colour indicates increases or decreases with statistical significance.
Authors: we improved the formatting of tables as recommended by the reviewer. We changed the foot-notes in tables 2 and 3, in the new version of the manuscript as follows: Shaded lines show cellular proteins for which significant abundance changes were detected: : the two highlighted proteins showed increased abundance in the msr mutant (Table S5)” and"Shaded lines show cellular proteins for which significant abundance changes were detected: the four highlighted proteins showed decreased abundance in the msr mutant (Table S6)"
Reviewer:L365/366 – what is the mechanisms (please see above), and how would MetO residue content be ‘regulated’ ( repaired?)
Authors: Our results suggest that some extracellular proteins could be Msr substrates. If there are Msr substrates, these proteins are oxidized and repaired in the cytoplasm before their exportation. We proposed this explanation in the discussion section.Concerning the terms regulated vs repaired, if the proteins are Msr substrates, they are repaired by Msr. In all cases, their Met(O) content are regulated by Msr, either directly or indirectly.
Reviewer: L406 – please add reference after ‘during the stationary phase.’ as hat data is not shown
Authors: we added the reference (27) as recommended by the reviewer.
Reviewer: L437/438 – what is meant by succinate pathway? If this is the part of the TCA cycle.
Authors: The succinate pathway is schematized in Fig. 5 and is the part of the reductive TCA cycle. We specified it in the legendof figure 5 in the new version of the manuscript.
Reviewer: L438 ff – How is NADH balance maintained when acetate production increases?
Authors: NADH balance is maintained through the lactate pathway since the production of lactate (in mol per mol of glucose consumed) is not changed in mutant strains (Table 1 and text). However, maintaining redox balance through the lactate pathway when acetate production increases requires higher rate of glucose consumption. As a results, the final biomass of mutant strains is lower.
Reviewer: L447/448 – could this be linked to the ATP synthase deficiency?
Authors: This could be linked to the ATP synthase deficiency. We added this possibility in the new version of the manuscript, and in the figure 5.
Reviewer: L455/456 – this needs a better explanation ( please see above), this interpretation is very speculative.
Authors: we have made the following changes in the manuscript: “All these virulence factors contain several Met(O), are highly produced at the end of growth [52], and are exported in an unfolded form [53]. Msr are known to preferentially reduce unfolded oxidized proteins [17], suggesting that these virulence factors, especially enterotoxin Hbl, could be repaired by Msr before their secretion. Whether Msr-dependent Met oxidation regulates exotoxin secretion, structure and activity remains an open question”.
Reviewer: L467/468 – This is a major issue: It is not clear how it is chemically possible that one stereoisomer of methionine would form preferentially during chemically induced oxidative damage. I am not aware of a mechanism that would explain such a preferential formation. Please provide a chemically plausible explanation, details relating to the nature of oxidation on the specific substrate proteins that corroborate this suggestion, or remove this part of the explanation. It is, in as far as I am aware, an unresolved question why in different organisms, MsrA or MsrB proteins can have key roles for cell health, while the enzyme converting the other MetO stereoisomer is absent or has a less important role. My personal view on this is that it is likely related to the protein substrates that are being repaired, where on key proteins due to the particular environment of the methionine residue R- and S-MetO may form preferentially.
Authors: We thank the reviewer for providing its point of view. We removed the explanation concerning the affinity of Msr for R-and S-Meto stereoisomers
Reviewer 2 Report
We provided a point-by-point response to the reviewers's comments as follows :
Reviewer 2: Line 95: to avoid confusion with the acronym indicating the synthesis phase of the cell cycle, the stationary growth phase should be indicated by a letter different than “S”. Maybe “St” ?
Authors: We used the letter” S” for stationary growth phase in several previous studies, and we prefer to keep this abbreviation to avoid any confusion with our previous works. We believe there will be no confusion with the synthesis phase of the cell cycle, at least for microbiologists.
Revewer 2: fine
Reviewer 2:
Line 207: "with a theoretical molecular weight of 20 475 Da." --> "with a theoretical molecular mass of 20 475 Da."
Authors: We replaced “weight” by “mass”.
Reviewer 2: very good
Reviewer 2:
Line 208: "This protein shares an MsrA domain with the bifunctional MsrAB." The domain is not "shared", being MsrA and MsrAB two different polypeptides. The degree of homology and identity of MsrA with the MsrA domain of MsrAB must be given.
Authors: We rewrote the sentence, as recommended by the reviewer and indicated the 64% of identity between the MsrA domains of MsrA and MsrAB in the new version of the manuscript.
Reviewer 2: good
Reviewer 2:
Lines 210-217. Following this part is difficult as it switches back and forth from aerobiosis to anaerobiosis. Fig 1S should be combined with Fig 1 in the main text to allow immediate comparison of Msr expression levels under the two growing conditions.
Authors: We appreciate the suggestion, but some of the results presented in Fig S1 were previously published. We therefore cannot put them back in the main text.
Reviewer 2: why not, it’s your results and it is not a repetition when added to the new results on growth in anaerobiosis.
Reviewer 2:
Lines 226-231. The most critical part of the present manuscript is the quantitative conclusions about levels of Msr expression. Accurate protein quantitation by proteomics is difficult: in this respect, the weakness of the present manuscript is in the lack of a more direct quantification of Msr expression, by enzymatic assay or at least Western blotting. After such a sophisticate proteomics approach what could only be concluded is that “… both Msr proteins are expressed at low abundance, and that their levels are reduced when cells are grown in anaerobic conditions compared to aerobic growth conditions.”
Authors: We quantified msr expression by qRT-PCR. Quantify Msr production by enzymatic assay under anaerobic conditions is extremely difficult. Western blot analysis requires to have antibodies against these proteins. There is no longer a need to validate proteomic studies with biochemical studies, the methodology being mature and widely accepted (see the editorial doi: 10.1074/mcp.E113.031658). Anyway, regardless of the amount of Msrs, our results clearly demonstrate their roles in anaerobic fermentative metabolism of B. cereus.
Reviewer 2: quantification of enzymatic activity in bacteria grown under anaerobiosis should not be more difficult than quantifying under aerobiosis. And then, everybody can do easy things. The concept of research is to try to do difficult, and extremely difficult things. I frankly dissent from the authors on the lack of need for validation of quantitative proteomic studies. Yes, this “mature technique” has been induced to mature under very “domesticated”, controlled conditions of well-established experimental setups, by resorting to isotope labelling and internal standards. Then, the validity of these results was confidently extrapolated to all other cell systems. Yet, this is quite arbitrary, because different proteins may behave very differently due to all sort of problems beginning from protein extraction, protease fragmentation by proteases, post-translational modification and artefactual modifications during sample manipulation. Nevertheless, again, there is not much at stake here to be overkill on this issue.
Reviewer 2:
Lines 254-266: the significance of statistical analysis, where present, must be fully reported for data in Table 1, not only the result of statistical comparison for acetate production.
Similarly, as there may seem to be a significant difference in the final biomass among the various strains (Fig 2), the full statistical comparison should be provided.
Authors: As recommended by both reviewers, we modified Table I and Fig. 2 to show the significance of the data.
Reviewer 2: good
Reviewer 2:
Lines 276-280: description of the difference in MetO levels among strains is inconsistent with what presented in the graph, probably due to a mistake in labelling the histograms.
Authors: there was indeed a mistake in labelling the histograms. We corrected these mistakes, and there is no longer inconsistency between description of the data and fig.3.
Reviewer 2: ok
Reviewer 2:
Lines 284-285: “Taken together, these data show that MsrA is a major contributor to the reduction of cellular protein-bound Met(O) under anaerobiosis, whereas MsrAB appears to play a minor role in this process.” Again, unless colors have been exchanged by mistake in Fig. 3, these statements are in conflict with the graph.
Authors: as indicated in our answers here-above, we corrected figure 3, and there is no longer inconsistency between description of the data and figure 3
Reviewer 2: ok
Reviewer 2:
From Line 297: Identification of Msr substrate proteins. There are several issues in this part. First, it is impossible for the reader to draw the same conclusions as the authors did from the raw data. Table 2 is supposed to give the peptides whose Met(O) content changed more than 1.5 times in any given mutant strain with respect to WT.
Authors: Table 2 reports the values of log2 FC, and not the values of FC. The Met(O) content is thus significantly changes when log2 FC>0.58.
Reviewer 2: Of course, this was clear. Still there is no need to use the binary logarithm of FC just because the software does so. Fold change would be perfectly fine.
Reviewer 2:
Take for instance the putative cytoplasmic protein Gls24 (the peptide VEIAPEVIEVIAGIAAAEVEGVAAMR, because there are other two peptides on which the authors do not comment). What are the numbers in Table 1S under the column “Met(O)”? in WT all the values are zero. In MsrA mutant, LE phase, the average of the three samples gives 17.7. But what does this number mean? Is it the quantity of peptide detected with the single Met in the peptide oxidised to Met(O)?. But how is then the “fold change” calculated? As “fold change” is the ratio between the final quantity and the initial quantity, it will always be zero if the starting value is zero!
Authors : Only the peptides carrying Met (O) whose rate changes significantly are reported in Table 2. Table S1 reports the number of peptides containing oxidized methionine in the cellular proteome. Fold-changes were determined using the DEP package in R, as reported in the material and method. This R package allows to transform the row data, and then to calculate the fold-changes. Details concerning the DEP package was described at https://www.bioconductor.org . In brief, to be analyzed, row data must be normalized and missing values (i.e values =0) must be filtered and imputed. Missing values are usually representative of the detection limit of the instrumentation. A missing value is a problem in a relative comparison experiment, even if a peptide is quantified in one of the two conditions. It is not possible to calculate a ratio if one of the two values is missing. Imputation of missing values is therefore performed by replacing them with estimated ones, usually representing detection limit. There are multiple ways to perform data imputation, and we used the MNAR method proposed in DEP. Then, fold changes can be estimated even if a peptide is not detected in one of the two strains. All proteomics bioinformatic tools used such transformation of raw data. This is well explained in the article of Aguilan et al (https://doi.org/10.1039/D0MO00087F)
Reviewer 2: Imputation of missing values is arbitrary and can lead to big mistakes. Again, overconfidence on in silico studies not validated by biochemical analysis.
Reviewer 2:
Then why give the fold change as the logarithm base 2? The tables must report plain “fold change”, not the logarithm, after explanation of how it is calculated.
Authors: Results obtained from DEP analysis are given as the logarithm base 2, and presenting the results in this form is fairly standard. In addition, if someone wanted to redo the calculations, they could directly compare their results to those shown in the article.
Reviewer 2:
There is no need to use the binary logarithm of FC just because the software does so. Fold change would be perfectly fine
Reviewer 2:
But no matter how the data have been elaborated, the only conclusion that is drawn is that only a couple of proteins among several dozens show an increase in their Met(O) content in the mutants. Some actually show a decrease. Of the two that display the increase, one is an unknown cytoplasmic protein. The other is sloppily defined as glucose 6 phosphate dehydrogenase in the text (Line 313) but it is actually phosphoglucose isomerase (sic)!
Authors : We have chosen to use strong statistical criteria, explaining why we found only 15 cellular and 21 extracellular proteins modified at the Met(O) level. Consequently, the significance of the results is very high.
We agree with the reviewer, there is an inconsistence between text and Table 2. We replaced in the text “glucose 6 phosphate dehydrogenase” by “glucose 6 phosphate isomerase” as indicated in the Table 2.
Reviewer 2: thanks for agreeing.
Reviewer 2:
Line 324 “a higher Met(O) content was detected for 32 Met residues compared to the levels present in WT.” please point to these 32 Met residues and their peptides (are they shown in Table 2?). Explain how was the increase in Met(O) content evaluated.
Authors: The Met residues are indicated in bold in column 6 of Tables 2 and 3 in the new version of the manuscript. We added foot-notes in Table 2, to facilitate Table reading. Increase of Met(O) level was evaluated using the DEP package as described in the material and method section.
Reviewer 2:
Line 325: “The abundance levels for the corresponding 13 proteins was unchanged (Table S4).” Make a separate list of the 13 proteins with a clear indication of the increase of Met oxidation that the authors can extrapolate from the raw data. As it stands, the presentation of the results is incomprehensible.
Authors: Table 2 lists the 15 proteins, with significant Met(O) level changes. Among them, 13 proteins did no shown abundance level changes, and 2 have shown abundance levels changes, as reported in Table S4. These two proteins are in grey in Table 2. We added a footnote in Table 2, explaining why these two proteins are in grey, while the 13 other proteins are in white.
Reviewer 2:
Line 315-318: How is it explained that all flagella proteins end up being more reduced in the mutants than in the WT (Table 3)? If this is because flagella proteins are also less expressed in the mutants, then this means that data in Table 2 and 3 are even more difficult to understand. In fact, it was assumed that the “fold change” in oxidation/reduction of Met in the proteins listed in the Tables was already normalized over protein abundance. Please explain.
Authors: We indicated in the text “Both the Met(O) level and overall abundance of four extracellular proteins was decreased in mutant strains compared to WT (Table 3). Among these proteins, three (flagellins FlaA, FlaB and FlaC) are components of the flagellar apparatus”. This suggests that the decrease Met(O) level of flagella protein is probably due to the decrease of their abundance. This point is discussed in the discussion section. Like in Table 2, we indicated in Table 3, proteins for which both Met(O) level and abundance level were significantly changes. These proteins are indicated in grey, and among these proteins are the flagellin proteins.
Reviewer 2:
Lack of oxidised Met in WT would point at an important role of “B” type of Msr, as MsrA alone would not be able to reduce the R diastereomer. So the role of MsrAB should not be underestimated, as it would appear form the conclusions of the paper. In addition, it appears that the presence of only MsrA and MsrAB in B. cereus was established on the basis of in silico studies. Can the authors exclude the existence of other Msr activities? Enzymatic assay of double mutant crude extracts could give some hints.
Authors: We did not underestimate the role of MsrAB. We only indicated that the role of MsrAB is less important in anaerobic fermentative metabolism than the role of MsrA. However, both the two proteins are important in regulating the Met(O) level of some virulence factors. The presence of MsrA and MsrAB was not established on the sole basis of in silico studies. We detected the mRNA coding for these two proteins, and we detected these two proteins in cellular proteome of B. cereus, using MS. We cannot exclude the existence of other Msr activities. However, the effects that we observed are only due to the absence of MsrA or/and MsrAB since we deleted the gene coding for these Msr, and showed that complementation of mutant strains restore the wild-type phenotype.
Reviewer 2:
Detecting mRNA encoding for known Msrs is finding what was already expected on the basis of in silico studies.
Author Response
We provided a point-by-point response to the reviewers's comments as follows :
Reviewer 2: Line 95: to avoid confusion with the acronym indicating the synthesis phase of the cell cycle, the stationary growth phase should be indicated by a letter different than “S”. Maybe “St” ?
Authors: We used the letter” S” for stationary growth phase in several previous studies, and we prefer to keep this abbreviation to avoid any confusion with our previous works. We believe there will be no confusion with the synthesis phase of the cell cycle, at least for microbiologists.
Reviewer 2: fine
Authors : We thanks the reviewer for her/his positive comments
Reviewer 2:
Line 207: "with a theoretical molecular weight of 20 475 Da." --> "with a theoretical molecular mass of 20 475 Da."
Authors: We replaced “weight” by “mass”.
Reviewer 2: very good
Authors : We thanks the reviewer for her/his positive comments
Reviewer 2:
Line 208: "This protein shares an MsrA domain with the bifunctional MsrAB." The domain is not "shared", being MsrA and MsrAB two different polypeptides. The degree of homology and identity of MsrA with the MsrA domain of MsrAB must be given.
Authors: We rewrote the sentence, as recommended by the reviewer and indicated the 64% of identity between the MsrA domains of MsrA and MsrAB in the new version of the manuscript.
Reviewer 2: good
Authors: We thanks the reviewer for her/his positive comments
Reviewer 2:
Lines 210-217. Following this part is difficult as it switches back and forth from aerobiosis to anaerobiosis. Fig 1S should be combined with Fig 1 in the main text to allow immediate comparison of Msr expression levels under the two growing conditions.
Authors: We appreciate the suggestion, but some of the results presented in Fig S1 were previously published. We therefore cannot put them back in the main text.
Reviewer 2: why not, it’s your results and it is not a repetition when added to the new results on growth in anaerobiosis.
Authors: We can be accused of plagiarism even when using our own results: we have already experienced it.
Reviewer 2:
Lines 226-231. The most critical part of the present manuscript is the quantitative conclusions about levels of Msr expression. Accurate protein quantitation by proteomics is difficult: in this respect, the weakness of the present manuscript is in the lack of a more direct quantification of Msr expression, by enzymatic assay or at least Western blotting. After such a sophisticate proteomics approach what could only be concluded is that “… both Msr proteins are expressed at low abundance, and that their levels are reduced when cells are grown in anaerobic conditions compared to aerobic growth conditions.”
Authors: We quantified msr expression by qRT-PCR. Quantify Msr production by enzymatic assay under anaerobic conditions is extremely difficult. Western blot analysis requires to have antibodies against these proteins. There is no longer a need to validate proteomic studies with biochemical studies, the methodology being mature and widely accepted (see the editorial doi: 10.1074/mcp.E113.031658). Anyway, regardless of the amount of Msrs, our results clearly demonstrate their roles in anaerobic fermentative metabolism of B. cereus.
Reviewer 2: quantification of enzymatic activity in bacteria grown under anaerobiosis should not be more difficult than quantifying under aerobiosis. And then, everybody can do easy things. The concept of research is to try to do difficult, and extremely difficult things. I frankly dissent from the authors on the lack of need for validation of quantitative proteomic studies. Yes, this “mature technique” has been induced to mature under very “domesticated”, controlled conditions of well-established experimental setups, by resorting to isotope labelling and internal standards. Then, the validity of these results was confidently extrapolated to all other cell systems. Yet, this is quite arbitrary, because different proteins may behave very differently due to all sort of problems beginning from protein extraction, protease fragmentation by proteases, post-translational modification and artefactual modifications during sample manipulation. Nevertheless, again, there is not much at stake here to be overkill on this issue.
Authors: Working in an anaerobic chamber is more difficult than working on a conventional bench. Nevertheless, we agree that we could determine the in vitro activity of Msr activity in an anaerobic chamber. We are also convinced that it will be interesting. However, we are not convinced that these experiments are essential to our work, which clearly shows the in vivo activity of Msr.
Reviewer 2:
Lines 254-266: the significance of statistical analysis, where present, must be fully reported for data in Table 1, not only the result of statistical comparison for acetate production.
Similarly, as there may seem to be a significant difference in the final biomass among the various strains (Fig 2), the full statistical comparison should be provided.
Authors: As recommended by both reviewers, we modified Table I and Fig. 2 to show the significance of the data.
Reviewer 2: good
Authors: We thanks the reviewer for her/his positive comments
Reviewer 2:
Lines 276-280: description of the difference in MetO levels among strains is inconsistent with what presented in the graph, probably due to a mistake in labelling the histograms.
Authors: there was indeed a mistake in labelling the histograms. We corrected these mistakes, and there is no longer inconsistency between description of the data and fig.3.
Reviewer 2: ok
Authors: We thanks the reviewer for her/his positive comments
Reviewer 2:
Lines 284-285: “Taken together, these data show that MsrA is a major contributor to the reduction of cellular protein-bound Met(O) under anaerobiosis, whereas MsrAB appears to play a minor role in this process.” Again, unless colors have been exchanged by mistake in Fig. 3, these statements are in conflict with the graph.
Authors: as indicated in our answers here-above, we corrected figure 3, and there is no longer inconsistency between description of the data and figure 3
Reviewer 2: ok
Authors: We thanks the reviewer for her/his positive comments
Reviewer 2:
From Line 297: Identification of Msr substrate proteins. There are several issues in this part. First, it is impossible for the reader to draw the same conclusions as the authors did from the raw data. Table 2 is supposed to give the peptides whose Met(O) content changed more than 1.5 times in any given mutant strain with respect to WT.
Authors: Table 2 reports the values of log2 FC, and not the values of FC. The Met(O) content is thus significantly changes when log2 FC>0.58.
Reviewer 2: Of course, this was clear. Still there is no need to use the binary logarithm of FC just because the software does so. Fold change would be perfectly fine.
Authors 2 : We agree that fold-change and not log2 FC would be used. However, it is the log2 which is generally given in the articles. Therefore, any comparison with other works will be facilitated if we keep the log2 FC.
Reviewer 2:
Take for instance the putative cytoplasmic protein Gls24 (the peptide VEIAPEVIEVIAGIAAAEVEGVAAMR, because there are other two peptides on which the authors do not comment). What are the numbers in Table 1S under the column “Met(O)”? in WT all the values are zero. In MsrA mutant, LE phase, the average of the three samples gives 17.7. But what does this number mean? Is it the quantity of peptide detected with the single Met in the peptide oxidised to Met(O)?. But how is then the “fold change” calculated? As “fold change” is the ratio between the final quantity and the initial quantity, it will always be zero if the starting value is zero!
Authors : Only the peptides carrying Met (O) whose rate changes significantly are reported in Table 2. Table S1 reports the number of peptides containing oxidized methionine in the cellular proteome. Fold-changes were determined using the DEP package in R, as reported in the material and method. This R package allows to transform the row data, and then to calculate the fold-changes. Details concerning the DEP package was described at https://www.bioconductor.org . In brief, to be analyzed, row data must be normalized and missing values (i.e values =0) must be filtered and imputed. Missing values are usually representative of the detection limit of the instrumentation. A missing value is a problem in a relative comparison experiment, even if a peptide is quantified in one of the two conditions. It is not possible to calculate a ratio if one of the two values is missing. Imputation of missing values is therefore performed by replacing them with estimated ones, usually representing detection limit. There are multiple ways to perform data imputation, and we used the MNAR method proposed in DEP. Then, fold changes can be estimated even if a peptide is not detected in one of the two strains. All proteomics bioinformatic tools used such transformation of raw data. This is well explained in the article of Aguilan et al (https://doi.org/10.1039/D0MO00087F)
Reviewer 2: Imputation of missing values is arbitrary and can lead to big mistakes. Again, overconfidence on in silico studies not validated by biochemical analysis.
Author 2: we agree that the calculated FC are probably not the same as those obtained by biochemical experiments. However, if a Met is found to be more oxidized by proteomics, it is unlikely that the reverse will be found with biochemical tests.
Reviewer 2:
Then why give the fold change as the logarithm base 2? The tables must report plain “fold change”, not the logarithm, after explanation of how it is calculated.
Authors: Results obtained from DEP analysis are given as the logarithm base 2, and presenting the results in this form is fairly standard. In addition, if someone wanted to redo the calculations, they could directly compare their results to those shown in the article.
Reviewer 2:
There is no need to use the binary logarithm of FC just because the software does so. Fold change would be perfectly fine
Authors 2 : We agree that fold-change and not log2 FC would be used. However, it is the log2 which is generally given in the articles. Therefore, any comparison will be facilitated if we keep the log2 FC.
Reviewer 2:
But no matter how the data have been elaborated, the only conclusion that is drawn is that only a couple of proteins among several dozens show an increase in their Met(O) content in the mutants. Some actually show a decrease. Of the two that display the increase, one is an unknown cytoplasmic protein. The other is sloppily defined as glucose 6 phosphate dehydrogenase in the text (Line 313) but it is actually phosphoglucose isomerase (sic)!
Authors : We have chosen to use strong statistical criteria, explaining why we found only 15 cellular and 21 extracellular proteins modified at the Met(O) level. Consequently, the significance of the results is very high.
We agree with the reviewer, there is an inconsistence between text and Table 2. We replaced in the text “glucose 6 phosphate dehydrogenase” by “glucose 6 phosphate isomerase” as indicated in the Table 2.
Reviewer 2: thanks for agreeing.
Authors: We thanks the reviewer for her/his positive comments
Reviewer 2:
Line 324 “a higher Met(O) content was detected for 32 Met residues compared to the levels present in WT.” please point to these 32 Met residues and their peptides (are they shown in Table 2?). Explain how was the increase in Met(O) content evaluated.
Authors: The Met residues are indicated in bold in column 6 of Tables 2 and 3 in the new version of the manuscript. We added foot-notes in Table 2, to facilitate Table reading. Increase of Met(O) level was evaluated using the DEP package as described in the material and method section.
Reviewer 2:
Line 325: “The abundance levels for the corresponding 13 proteins was unchanged (Table S4).” Make a separate list of the 13 proteins with a clear indication of the increase of Met oxidation that the authors can extrapolate from the raw data. As it stands, the presentation of the results is incomprehensible.
Authors: Table 2 lists the 15 proteins, with significant Met(O) level changes. Among them, 13 proteins did no shown abundance level changes, and 2 have shown abundance levels changes, as reported in Table S4. These two proteins are in grey in Table 2. We added a footnote in Table 2, explaining why these two proteins are in grey, while the 13 other proteins are in white.
Reviewer 2:
Line 315-318: How is it explained that all flagella proteins end up being more reduced in the mutants than in the WT (Table 3)? If this is because flagella proteins are also less expressed in the mutants, then this means that data in Table 2 and 3 are even more difficult to understand. In fact, it was assumed that the “fold change” in oxidation/reduction of Met in the proteins listed in the Tables was already normalized over protein abundance. Please explain.
Authors: We indicated in the text “Both the Met(O) level and overall abundance of four extracellular proteins was decreased in mutant strains compared to WT (Table 3). Among these proteins, three (flagellins FlaA, FlaB and FlaC) are components of the flagellar apparatus”. This suggests that the decrease Met(O) level of flagella protein is probably due to the decrease of their abundance. This point is discussed in the discussion section. Like in Table 2, we indicated in Table 3, proteins for which both Met(O) level and abundance level were significantly changes. These proteins are indicated in grey, and among these proteins are the flagellin proteins.
Reviewer 2:
Lack of oxidised Met in WT would point at an important role of “B” type of Msr, as MsrA alone would not be able to reduce the R diastereomer. So the role of MsrAB should not be underestimated, as it would appear form the conclusions of the paper. In addition, it appears that the presence of only MsrA and MsrAB in B. cereus was established on the basis of in silico studies. Can the authors exclude the existence of other Msr activities? Enzymatic assay of double mutant crude extracts could give some hints.
Authors: We did not underestimate the role of MsrAB. We only indicated that the role of MsrAB is less important in anaerobic fermentative metabolism than the role of MsrA. However, both the two proteins are important in regulating the Met(O) level of some virulence factors. The presence of MsrA and MsrAB was not established on the sole basis of in silico studies. We detected the mRNA coding for these two proteins, and we detected these two proteins in cellular proteome of B. cereus, using MS. We cannot exclude the existence of other Msr activities. However, the effects that we observed are only due to the absence of MsrA or/and MsrAB since we deleted the gene coding for these Msr, and showed that complementation of mutant strains restore the wild-type phenotype.
Reviewer 2:
Detecting mRNA encoding for known Msrs is finding what was already expected on the basis of in silico studies.
Authors: Detecting mRNA makes it possible to show that genes are expressed and therefore that proteins are produced. Usually bacteria do not transcribe genes for nothing. Post-translational regulations are minor compared to post-transcriptional regulations in bacteria.
Round 3
Reviewer 1 Report
Thank you for comprehensively addressing my concerns. I only have one final change that I would like to see for the manuscript, and this concerns the use of letters in the figures to indicate statistical significance.
Regardless of whether stars or letters are used to indicate statistical significance, it is essential that the meaning of each letter/ set of stars is clearly defined in the figure legend. Simply stating that for all elements with letters P<0.05 is not sufficient. Could the figure legends please be amended to have information in the following or a similar format
a: p <0.05; b: p<0.01 etc etc.
particularly where combinations of letters are used it is essential to ensure that the meaning of these combinations is clear ( e.g. meaning of ab vs a alone of b alone).
Please also add what comparisons were undertaken - were statistically significant changes evaluated relative to the Wt in the same group, or were comparisons between the groups/ between mutant strains in each group also undertaken?
Author Response
Thank you for comprehensively addressing my concerns. I only have one final change that I would like to see for the manuscript, and this concerns the use of letters in the figures to indicate statistical significance.
Authors: We appreciated the positive overall evaluation of our work. Our answers to the reviewers’ comments are given below.
Reviewer 1: Regardless of whether stars or letters are used to indicate statistical significance, it is essential that the meaning of each letter/ set of stars is clearly defined in the figure legend. Simply stating that for all elements with letters P<0.05 is not sufficient. Could the figure legends please be amended to have information in the following or a similar format
a: p <0.05; b: p<0.01 etc etc.
particularly where combinations of letters are used it is essential to ensure that the meaning of these combinations is clear ( e.g. meaning of ab vs a alone of b alone).
Please also add what comparisons were undertaken - were statistically significant changes evaluated relative to the Wt in the same group, or were comparisons between the groups/ between mutant strains in each group also undertaken?
Authors: Differences between mean values are indicated using letters with the following conventional standard: lowercase letters for p<0.05, and uppercase letters for p<0.01. Therefore, different lowercase letters do no denote different p-values. Different lowercase letters denote significant difference between the values, with p<0.05. Values denoted by “a” are significantly different that values denoted by (b) with p<0.05. Because there is one letter in common, values denoted by “ab” are not significantly different that values denoted by “a”, and not significantly different that values denoted by “b”. We have changed the legend of the figures to make this clearer. We also added in the figure legends of the new version of the manuscript that multiple comparisons were realized to evaluate statistical differences.
We replaced in the legend of figure 1: “ Data were evaluated by a two-way ANOVA followed by Tukey's post hoc analysis. Different letters indicate a significant difference (p≤0.05) between strains at each growth phase » by: “Data denoted by a common letter are not significantly different. Data denoted by different letters indicated a significant difference (two-way ANOVA followed by Tukey's multiple comparison post hoc analysis, p≤0.05).”
We replaced in the legend of figure 3:” Data were evaluated by a two-way ANOVA followed by Tukey's post hoc analysis. Different letters in each panel indicate a significant difference (p≤0.05) between strains at each growth phase” by “Within each panel, data correspond to the mean ± SD of three biological replicates. Data denoted by a common letter are not significantly different. Data designed by different letters indicated a significant difference (two-way ANOVA followed by Tukey's multiple comparison post hoc analysis, p≤0.05)”.